# LATENT 3D GRAPH DIFFUSION

**Yuning You**[1]**, Ruida Zhou**[2]**, Jiwoong Park**[1]**, Haotian Xu**[1]**,
Chao Tian**[1]**, Zhangyang Wang**[3]**, Yang Shen**[1]

[1]Texas A&M University, [2]University of California, Los Angeles, [3]University of Texas at Austin
{yuning.you,jiwoong.park,hx105,chao.tian,yshen}@tamu.edu,
ruida@g.ucla.edu, atlaswang@utexas.edu

## ABSTRACT

Generating 3D graphs of *symmetry-group equivariance* is of intriguing potential in broad applications from machine vision to molecular discovery. Emerging approaches adopt diffusion generative models (DGMs) with proper re-engineering to capture 3D graph distributions. In this paper, we raise an orthogonal and fundamental question of *in what (latent) space we should diffuse 3D graphs*. ❶ We motivate the study with theoretical analysis showing that the performance bound of 3D graph diffusion can be improved in a latent space versus the original space, provided that the latent space is of (i) low dimensionality yet (ii) high quality (i.e., low reconstruction error) and DGMs have (iii) symmetry preservation as an inductive bias. ❷ Guided by the theoretical guidelines, we propose to perform 3D graph diffusion in a low-dimensional latent space, which is learned through cascaded 2D–3D graph autoencoders for low-error reconstruction and symmetry-group invariance. The overall pipeline is dubbed **latent 3D graph diffusion**. ❸ Motivated by applications in molecular discovery, we further extend latent 3D graph diffusion to conditional generation given SE(3)-invariant attributes or equivariant 3D objects. ❹ We also demonstrate empirically that out-of-distribution conditional generation can be further improved by regularizing the latent space via graph self-supervised learning. We validate through comprehensive experiments that our method generates 3D molecules of higher validity / drug-likeliness and comparable or better conformations / energetics, while being an order of magnitude faster in training. Codes are released at https://github.com/Shen-Lab/LDM-3DG.

## 1 INTRODUCTION

Generative AI is shifting the paradigms in broad applications, with its intriguing potential to simulate various real-world data (Ho et al., 2020; Song et al., 2020). In this paper, the research focus is *3D graph generation* (see Fig. 1 and Sec. 2 for definition), which is of significant needs in scientific discovery ranging from biomolecules (Morris & Lim-Wilby, 2008) to spatial transcriptomics (Burgess, 2019). The emerging approaches (Hoogeboom et al., 2022; Wu et al., 2022; Vignac et al., 2023; Morehead & Cheng, 2023) adopt *diffusion generative models* (DGMs) (Cao et al., 2022; Croitoru et al., 2023; Yang et al., 2022), a class of generative AI that witnessed startling empirical successes in generating Euclidean data, with additional architectural redesigns to preserve *permutation and SE(3) in/equivariance* for non-Euclidean 3D graphs.

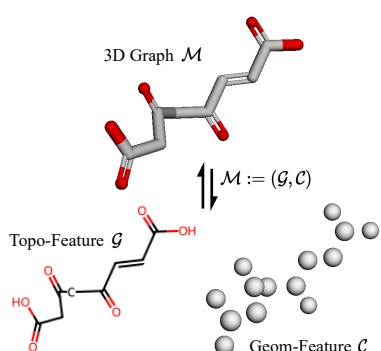

**Figure 1:** 3D graph is composed of the topological (connectivities) and geometric features (spatial coordinates).

Inspired by the seminal works (Hoogeboom et al., 2022; Wu et al., 2022; Vignac et al., 2023; Morehead & Cheng, 2023) on how to build 3D graph DGMs, this paper drives the exploration by asking an orthogonal, under-investigated yet fundamental question: *In what space should we diffuse 3D graphs?* Typically DGMs are built directly in the product space of 3D graph topology (i.e. nodes and edges) and geometry (i.e. coordinates). Such space is an

intuitive choice where symmetry structures exist, but can be arduous for DGMs to capture resulting distributions (De Bortoli, 2022; Liu et al., 2023). The reason is that real-world data usually distribute on constrained, *low-dimensional* manifolds (De Bortoli, 2022). For instance, arbitrarily perturbing atom types or coordinates could lead to stereochemically invalid molecules (Mislow, 2002) and randomly distorting expression profiles would result in biologically aberrant transcriptomics (Marsh et al., 2022). We hypothesize that a more *compact*, yet *informative* and *symmetry-preserved* space w.r.t. such manifold can benefit 3D graph diffusion generation in both quality and speed.

**Contributions.**

❶ *What (latent) space for 3D graph diffusion?* (Sec. 3.2) We provide a motivational analysis showing that 3D graph diffusion performance bound is related to (i) dimensionality of latent embeddings, (ii) quality of latent spaces w.r.t. distributions, and (iii) symmetry preservation of DGMs. Our analysis indicates that, the performance for 3D graphs can be improved in a latent space, compared to that in the original space if the latent space is constructed with (i) low dimensionality and (ii) low reconstruction error, which present a trade-off, while (iii) preserving group symmetry.

❷ *How to construct a compact and informative space for 3D graph diffusion with symmetry preserved?* (Sec. 3.1) Compared to Euclidean-structured data (Rombach et al., 2022; Jing et al., 2022), 3D graph data are much more challenging to "parametrize" due to the symmetry groups of permutation and SE(3) transformations (Hoogeboom et al., 2022; Wu et al., 2022) and the abstraction of data of diverse natures (You et al., 2020a; 2021). Our solution is to learn a low-dimension latent space parametrization in a data-driven manner, by pretraining a 3D graph autoencoder (AE). We innovate in building the AE architecture strategically on the decomposed topological and geometric features such that the symmetry constraints of permutation and SE(3) transformations are disentangled and properly tackled in separate but cascaded AE models. DGM is then trained in the resulting latent space to model distributions, and the overall pipeline is dubbed **latent 3D graph diffusion**.

❸ *How to introduce latent diffusion for conditional 3D graph generation with additional symmetry structures w.r.t. conditions?* (Sec. 3.3) Through the motivating example of molecular discovery, we further extend our pipeline to the more challenging conditional generation, where the generated 3D graphs need to be invariant to scalar attributes or equivariant to 3D objects. We achieve this by redesigning 3D graph AEs and diffusion with conditional in/equivariance preserved.

❹ *How to regularize latent space for better "generalizability" of latent 3D graph diffusion?* We empirically observe that the out-of-distribution robustness can be improved with appropriate regularization of the latent space via graph self-supervised learning; and the extent of semantic alignment (Kwon et al., 2022) in the latent space is indicative of generation quality.

Experimental results (Sec. 4) demonstrate that, our method is capable of generating 3D molecular graphs of higher validity / drug-likeliness and comparable conformations / energetics compared to state-of-the-art competitors, while being at least an order of magnitude faster in diffusion training. The advantage in speed can increase as the molecule size / complexity increases.

## 2 RELATED WORKS

**3D graph generation.** A 3D graph $\mathcal{M} := (\mathcal{G}, \mathcal{C}) \in \mathbb{M}$ is composed of topology $\mathcal{G}$ and geometry $\mathcal{C}$ (Barthélemy, 2011) as shown in Fig. 1. One inspiring example is molecules (Hoogeboom et al., 2022; Wang et al., 2022) whose topology is $\mathcal{G} = (\mathcal{A}, \mathcal{B})$, where $a \in \mathcal{A}$ (nodes) contains information of atom identities and features and $b \in \mathcal{B}$ (edges) of bond types, and geometry is $c \in \mathcal{C}$ denoting the 3D coordinates of a node. Therefore, the aim of 3D graph generation is to learn the underlying distribution $p_\theta$ parametrized by $\theta$, of the set of observed samples $\mathcal{D} = \{\mathcal{M}_1, \mathcal{M}_2, ...\}, \mathcal{M}_i \sim p_{\text{data}}(\mathcal{M})$ with the optimization $\min_\theta \ell(\mathcal{D}, \bar{\mathcal{D}})$, where $\bar{\mathcal{D}} = \{\bar{\mathcal{M}}_1, \bar{\mathcal{M}}_2, ...\}$ is the set of generated samples that $\bar{\mathcal{M}}_i \sim p_\theta(\mathcal{M})$, and $\ell(\cdot)$ is objective function. The emerging approaches adopt varied generative models to the 3D graph space $\mathbb{M}$, including autoregressive (Gebauer et al., 2019), normalizing flows (Satorras et al., 2021), and recently diffusion models (Hoogeboom et al., 2022; Wu et al., 2022; Vignac et al., 2023; Morehead & Cheng, 2023), with additional architectural redesigns to certify permutation and SE(3) in/equivariance.

**Diffusion models.** Diffusion generative models (DGMs) (Sohl-Dickstein et al., 2015; Ho et al., 2020; Song et al., 2020; Chen et al., 2022a) have received an explosion of interest recently, for their startling empirical successes in modeling distributions of Euclidean data, denoted as $p_{\text{data}}(\mathbf{x})$ in the $\mathbb{R}^D$ space with dimension $D$. The forward diffusion process perturbs data with noises in time $T$ through a stochastic differential equation $d\mathbf{x}^{(t)} = \mathbf{x}^{(t)}dt + \sqrt{2}d\mathbf{w}, \mathbf{x}^{(0)} \sim p_{\text{data}}(\mathbf{x}), \mathbf{x}^{(t)} \sim p^{(t)}(\mathbf{x})$

where $\mathbf{w}$ is the Wiener process (Ross, 1995), and the reverse reconstructs from noisy data via $d\bar{\mathbf{x}}^{(t)} = \left(\bar{\mathbf{x}}^{(t)} + 2\nabla_{\mathbf{x}} \ln p^{(T-t)}(\bar{\mathbf{x}}^{(t)})\right)dt + \sqrt{2}d\mathbf{w}, \bar{\mathbf{x}}^{(0)} \sim p^{(T)}(\mathbf{x})$. DGMs thus are targeted at training a score estimator $f_\theta : \mathbb{R}^D \times \mathbb{R}_{\geq 0} \to \mathbb{R}^D$ parametrized by $\theta$ with the optimization $\min_\theta \mathbb{E}_{p(t), p^{(t)}(\mathbf{x}^{(t)})} \| f_\theta(\mathbf{x}^{(t)}, t) - \nabla_{\mathbf{x}} \ln p^{(t)}(\mathbf{x}^{(t)}) \|^2$. In practice, the continuous processes are usually discretized into $S$ steps with step size $\frac{T}{S}$ (Sohl-Dickstein et al., 2015; Ho et al., 2020).

**Latent diffusion and autoencoders for structured data.** Recent work (Rombach et al., 2022; Yu et al., 2020) has shown improved generative modeling by exploring expressive generative models over the latent space of Euclidean data, which is parametrized with autoencoders (AEs). As a well-studied data-driven dimension reduction technique, AEs (Kramer, 1991; Kingma & Welling, 2013) have been recently explored for 2D topological graphs (Simonovsky & Komodakis, 2018; Jin et al., 2020; Tang et al., 2022; Gómez-Bombarelli et al., 2018; De Cao & Kipf, 2018; Lim et al., 2018; Woodward; Huang et al., 2022; Chen et al., 2022b) and point clouds (Anvekar et al., 2022; Zeng et al., 2022; Zhu et al., 2022a). The investigation on 3D graph AEs and their impact on 3D graph diffusion however remains open.

## 3 METHODS

We begin with the conclusion of theoretical analysis to elucidate the motivation behind the proposed **latent 3D graph diffusion** (Proposition 2 in Sec. 3.2), described informally as follows:

$$\text{3D Graph Diffusion Performance} \leq \text{Latent Space Reconstruction Quality}$$
$$+ \text{ Symmetry Preservation} \times \text{Data Dimensionality}.$$

One end of the spectrum is equivariant diffusion (e.g. (Hoogeboom et al., 2022; Wu et al., 2022)) in the original 3D graph space which results in the highest dimensionality but lossless reconstruction. We aim at better DGMs in the constructed latent space that balances reconstruction and dimensionality, while maintaining symmetry for better generation.

With the above guidelines, the proposed method comprises the following three components, with an overview depicted in Fig. 2(a). The pivotal one is the cascaded 2D–3D graph autoencoder that constructs a high-quality and lower-dimensional latent space in a data-driven manner, and is invariant to permutation and SE(3) transformations, where the diffusion model is built upon.

- *Pretrained 3D graph autoencoder* (AE) to map between non-Euclidean structures $\mathcal{M}$ and latent embeddings $\mathbf{z}$: $\mathbf{z} = \overrightarrow{h}_{\phi_1}(\mathcal{M}), \bar{\mathcal{M}} = \overleftarrow{h}_{\phi_2}(\mathbf{z})$. We build a qualified 3D graph AE via (i) decomposing features into topological and geometric views and (ii) auto-encoding, to bypass the optimization barrier on the complex data structures regarding symmetry (to achieve sufficiently low reconstruction errors versus non-cascaded AEs, Sec. 3.1 & Fig. 2(b)), while preserving permutation- and SE(3)-equivariance, respectively.

- *Diffusion generative model* (DGM) to capture data distributions $p_\theta(\mathbf{z})$ in the latent space. The implemented DGM in the pipeline is standardized as described in Sec. 2.

- *Conditional in/equi-variance* fed to DGM. In SE(3) invariant conditional generation, we featurize SE(3)-invariant conditions $\mathbf{x}_{\text{cond}}$ and overwrite the score estimator as $f_\theta(\mathbf{z}^{(t)}, \mathbf{x}_{\text{cond}}, t)$. We also extend the pipeline to SE(3)-equivariant generation conditioned on 3D objects (Sec. 3.3 & Fig. 3). Besides, we investigate how to improve "generalizability" of conditional generation via regularizing latent spaces with graph self-supervised learning (Fig. 2(c)).

The sampling procedure is cascaded with latent sampling $\bar{\mathbf{z}} \sim p_\theta$ and then decoding $\bar{\mathcal{M}} = \overleftarrow{h}_{\phi_2}(\bar{\mathbf{z}})$. Please refer to Algs. 1 & 2 for details.

### 3.1 CASCADED AUTO-ENCODING ON DECOMPOSED VIEWS OF 3D GRAPHS

Our preliminary efforts attempt to build a 3D graph AE directly in "*one shot*", that embeds and reconstructs topology and geometry simultaneously, via $\min_{\phi_1, \phi_2} \varepsilon\left(\overleftarrow{h}_{\phi_2}(\overrightarrow{h}_{\phi_1}(\mathcal{M})), \mathcal{M}\right)$. The permutation- and SE(3)-invariance is guaranteed in the objective functions $\varepsilon(\cdot)$ via graph matching (Yu et al., 2020) and superimposition algorithms (Kabsch, 1976) (see Append. B.1). However, substantial difficulty is encountered in optimizing the 3D graph AE with the intertwined symmetry constraints (of permutation and SE(3)) on such complex data structures (see Tab. 1 one-shot AE reconstruction performance on 3D molecules and Tab. 16 for more variants).

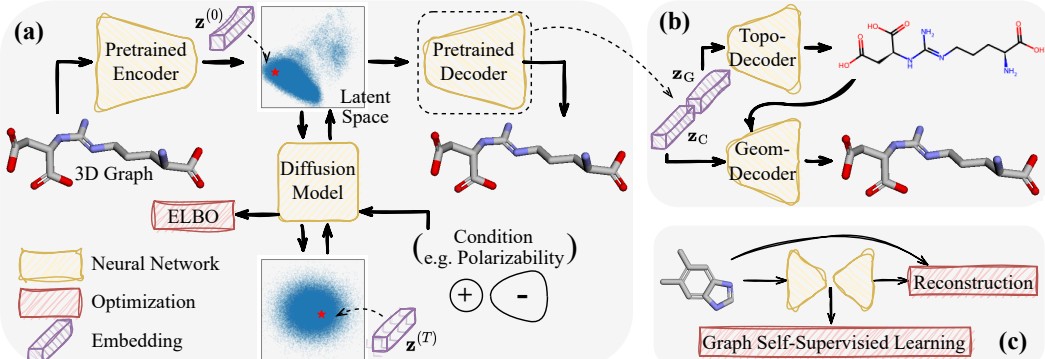

**Figure 2:** (a) Pipeline overview of self-supervised latent 3D graph diffusion where the latent space is (b) learned by an autoencoder architecture on the decomposed 3D graph topologies and geometries and can be (c) regularized by graph self-supervision through an auxiliary objective during autoencoder training.

**Cascaded AEs on decomposed 3D graph features with disentangled symmetry constraints.** Instead of auto-encoding the 3D graph features $\mathcal{M}$ straightforwardly in one shot, we propose to build AEs on the *decomposed* features of topology $\mathcal{G}$ and geometry $\mathcal{C}$ where the symmetry constraints are disentangled: topological AE are optimized with the permutation constraint and geometric AE with SE(3) (Fig. 2(b)). We are thus able to bypass the optimization barrier in a "divide and conquer" manner, mainly inspired by the sub-band coding theory

**Table 1:** Reconstruction performance of AE workflows on molecule data, evaluated with topological accuracy (Recon.), Tanimoto score (Tani.), and geometric root-mean-square error (RMSE).

| Methods | Recon.↑ | Tani.↑ | RMSE↓ |
|---|---|---|---|
| Random Init. | 0% | 9.50% | 1.86 |
| One-Shot AE | 0.80% | – | 1.80 |
| Cascaded AE | **79.57**% | **95.68**% | **0.69** |

(Vetterli, 1984): signals are divided into different frequency bands before/after encoding/decoding to be transmitted in the bandlimited system. The auto-encoding processes are formulated as:

$$\text{Encoding:} \quad \mathbf{z}_{\text{G}} = \overrightarrow{h}_{\phi_{1,\text{G}}}(\mathcal{G}), \quad \mathbf{z}_{\text{C}} = \overrightarrow{h}_{\phi_{1,\text{C}}}(\mathcal{C}), \quad \mathbf{z} = [\mathbf{z}_{\text{G}}; \mathbf{z}_{\text{C}}],$$

$$\text{Decoding:} \quad \bar{\mathcal{G}} = \overleftarrow{h}_{\phi_{2,\text{G}}}(\mathbf{z}_{\text{G}}), \quad \bar{\mathcal{C}} = \overleftarrow{h}_{\phi_{2,\text{C}}}(\bar{\mathcal{G}}, \mathbf{z}_{\text{C}}), \tag{1}$$

where the topological and geometric latent encoders $\overrightarrow{h}_{\phi_{1,\text{G}}}, \overrightarrow{h}_{\phi_{1,\text{C}}}$ are constructed to be permutation- and SE(3)-invariant, respectively, together with the invariant training objectives in optimization as $\min_{\phi_{1,\text{G}},\phi_{2,\text{G}}} \mathbb{E}_{p(\mathcal{G})} \varepsilon_{\text{G}} \left( \overleftarrow{h}_{\phi_{2,\text{G}}}(\overrightarrow{h}_{\phi_{1,\text{G}}}(\mathcal{G})), \mathcal{G} \right)$ (Jin et al., 2020) and $\min_{\phi_{1,\text{C}},\phi_{2,\text{C}}} \mathbb{E}_{p(\mathcal{G},\mathcal{C})} \varepsilon_{\text{C}} \left( \overleftarrow{h}_{\phi_{2,\text{C}}}(\mathcal{G}, \overrightarrow{h}_{\phi_{1,\text{C}}}(\mathcal{C})), \mathcal{C} \right)$ (Zhu et al., 2022a).

Note that the 3D graph cascaded AE is pretrained for once and utilized repetitively. We adopt architectures from (Jin et al., 2020) for the topological AE and (Zhu et al., 2022a) for the geometric (see Append. B.2 for details). Tab. 1 demonstrates its qualified reconstruction.

## 3.2 ANALYSIS OF LATENT 3D GRAPH DIFFUSION

We would like to understand how the choice of diffusion space impacts the generation quality of 3D graphs. We provide an analysis extended from prior works (Chen et al., 2022a; Lee et al., 2023; Liu et al., 2022c; Block et al., 2020; Lee et al., 2022) to the non-Euclidean space of 3D graphs, by further considering the symmetry structures via introducing symmetry-group equivariance.

Our reaching conclusions are (**i**) 3D graph diffusion performance could be related to data dimensionality (Propos. 1), revealing the motivation for why needing latent 3D graph diffusion; and (**ii**) performance could be improved if mappings to the lower-dimensional space of 3D graphs are appropriately constructed (Propos. 2), identifying the guidelines to building 3D graph AEs.

**Setup.** The featurization of a 3D graph $\mathcal{M} = ((\mathcal{A}, \mathcal{B}), \mathcal{C})$ of size $N$ is instantiated as $\mathbf{x}_{\text{M}} = (\mathbf{x}_{\text{A}}, \mathbf{x}_{\text{C}})$ where $\mathbf{x}_{\text{A}} \in \mathbb{R}^{N \times D_{\text{A}}}$ is the node feature matrix of dimension $D_{\text{A}}$ and $\mathbf{x}_{\text{C}} \in \mathbb{R}^{N \times 3}$ the coordinate matrix. The connection information can be later determined with certain domain rules as $\mathcal{B} = \text{rule}(\mathcal{A}, \mathcal{C})$. Such featurization is widely adopted (Hoogeboom et al., 2022; Wu et al., 2022; Morehead & Cheng, 2023) and therefore serves our analysis.

Different from Euclidean data assessing the DGM performance with the "flat" distribution discrepancy $\text{Dist}(p_\theta, p_{\text{data}})$, e.g., the $\ell$-infinity, total variation distance (Chen et al., 2022a) or Wasserstein

metric (De Bortoli, 2022), there exist symmetry structures in the 3D graph space. We characterize it by introducing the equivariance class $[\mathbf{x}_M]_{\Pi,\Omega} := \{\pi \circ \omega(\mathbf{x}_M) : \forall \pi \in \Pi, \omega \in \Omega\}$, where $\Pi, \Omega$ are the $N$-permutation and SE(3) groups, respectively, i.e., $\pi(\mathbf{x}_M) = (\pi\mathbf{x}_A, \pi\mathbf{x}_C), \omega(\mathbf{x}_M) = (\mathbf{x}_A, \mathbf{x}_C\omega), \forall \pi \in \Pi, \omega \in \Omega$. Thus, the assessment for 3D graphs is formulated with the distribution discrepancy on equivariance classes as $\mathrm{Dist}(\tilde{p}_\theta, \tilde{p}_{\mathrm{data}})$, where $\tilde{p}_\theta([\mathbf{x}_M]_{\Pi,\Omega}) = \mathrm{Pr}\{\mathbf{x}'_M : \mathbf{x}'_M \in [\mathbf{x}_M]_{\Pi,\Omega}, \mathbf{x}'_M \sim p_\theta\}$ and $\tilde{p}_{\mathrm{data}}([\mathbf{x}_M]_{\Pi,\Omega}) = \mathrm{Pr}\{\mathbf{x}'_M : \mathbf{x}'_M \in [\mathbf{x}_M]_{\Pi,\Omega}, \mathbf{x}'_M \sim p_{\mathrm{data}}\}$.

We first illustrate the potential factors relevant to diffusion performance in the *3D graph space*.

**Proposition 1.** (Performance bound of 3D graph diffusion is related to **feature dimensionality** of data and model inductive bias. See proof in Append. A.1) Assume DGM is trained to model $p_{\mathrm{data}}(\mathbf{x}_M)$ with $p_\theta(\mathbf{x}_M)$, $\forall t \geq 0$ the score $\nabla_{\mathbf{x}} \ln p^{(t)}$ is $L$-Lipschitz, for some $\eta > 0$ the moment $\mathbb{E}_{p_{\mathrm{data}}(\mathbf{x}_M)}\|\mathbf{x}_M\|^{2+\eta}$ is finite, and $\forall t \geq 0$ the score estimation error is bounded that $\mathbb{E}_{p^{(t)}(\mathbf{x}_M^{(t)})}\|f_\theta(\mathbf{x}_M^{(t)}, t) - \nabla_{\mathbf{x}} \ln p^{(t)}(\mathbf{x}_M^{(t)})\|^2 \leq \varepsilon_{\mathrm{score}}^2$. Denote the second moment $\mathsf{m} = \mathbb{E}_{p_{\mathrm{data}}(\mathbf{x}_M)}\|\mathbf{x}_M\|^2$ and suppose the DGM step size is 1. Then, it holds for the 3D graph DGM assessment (total variation distance or TV below):

$$\mathrm{TV}(\tilde{p}_\theta, \tilde{p}_{\mathrm{data}}) \lesssim \alpha(p_\theta, \Pi, \Omega)\left(\sqrt{\mathrm{KL}(p_{\mathrm{data}}\|\mathcal{N}_{D'})}e^{-T} + (L\sqrt{D'} + L\mathsf{m} + \varepsilon_{\mathrm{score}})\sqrt{T}\right), \quad (2)$$

where $\alpha(\cdot)$ depends on both the score estimator architecture and the symmetric groups, and $\mathcal{N}_{D'}$ is the normal distribution of dimension $D'$ that $D' = N \times (D_A + 3)$. □

However it is not necessary to model the distribution spanning the full $\mathbb{R}^{D'}$ space, if (i) the distribution is supported constrainedly on a low-dimensional manifold, a.k.a. the manifold hypothesis (De Bortoli, 2022; Fefferman et al., 2016) (which applies to molecules), and (ii) the manifold (Do Carmo & Flaherty Francis, 1992) can be parametrized. We next illustrate the factors relevant to 3D graph diffusion performance in the *latent space*.

**Proposition 2.** (3D graph diffusion could benefit from the **lower-dimensional latent space if appropriately constructed**. See proof in Append. A.2) Assume there existing mappings $\overrightarrow{h} : \mathbb{R}^{D'} \to \mathbb{R}^{D''}, \overleftarrow{h} : \mathbb{R}^{D''} \to \mathbb{R}^{D'}$ that $D'' < D'$ and $\overleftarrow{h}$ is injective. Assume DGM now is trained in $\mathbb{R}^{D''}$ to model $\overrightarrow{p}_{\mathrm{data}}(\mathbf{z}) = \mathrm{Pr}\{\mathbf{x}_M : \overrightarrow{h}(\mathbf{x}_M) = \mathbf{z}, \mathbf{x}_M \sim p_{\mathrm{data}}\}$ with $p_\theta(\mathbf{z})$, and it is evaluated in $\mathbb{R}^{D'}$ on $\overleftarrow{p}_\theta([\mathbf{x}_M]_{\Pi,\Omega}) = \mathrm{Pr}\{\mathbf{z} : \overleftarrow{h}(\mathbf{z}) \in [\mathbf{x}_M]_{\Pi,\Omega}, \mathbf{z} \sim p_\theta\}$ (as in Propos. 1), and the assumptions in Propos. 1 retain for the score estimator $f_\theta$ and mapping distribution. Then, it holds:

$$\mathrm{TV}(\overleftarrow{p}_\theta, \tilde{p}_{\mathrm{data}}) \lesssim \mathrm{TV}(\overleftrightarrow{\tilde{p}}_{\mathrm{data}}, \tilde{p}_{\mathrm{data}}) +$$
$$\bar{\alpha}(p_\theta, \overrightarrow{h}, \overleftarrow{h}, \Pi, \Omega)\left(\sqrt{\mathrm{KL}(\overrightarrow{p}_{\mathrm{data}}\|\mathcal{N}_{D''})}e^{-T} + (L\sqrt{D''} + L\mathsf{m} + \varepsilon_{\mathrm{score}})\sqrt{T}\right), \quad (3)$$

where $\overleftrightarrow{\tilde{p}}_{\mathrm{data}}([\mathbf{x}_M]_{\Pi,\Omega}) = \mathrm{Pr}\{\mathbf{x}'_M : \overleftarrow{h}(\overrightarrow{h}(\mathbf{x}'_M)) \in [\mathbf{x}_M]_{\Pi,\Omega}, \mathbf{x}'_M \sim p_{\mathrm{data}}\}$, and $\bar{\alpha}(\cdot)$ depends on both the latent diffusion architecture that $\bar{\alpha}(p_\theta, \overrightarrow{h}, \overleftarrow{h}, \Pi, \Omega) = \alpha(\overleftarrow{p}_\theta, \Pi, \Omega)$ if $\overleftrightarrow{p}_{\mathrm{data}} = p_{\mathrm{data}}$. □

Both latent space and DGMs influence the generation quality. By comparing the bounds in Ineq. (2) (original space) and Ineq. (3) (latent space), assuming that $\Pi$ and $\Omega$ equivariance is satisfied in both DGMs, the performance bound in the latent space could outperform that in the original space only if $D'' < D'$ (dimensionality) and $\mathrm{TV}(\overleftrightarrow{p}_{\mathrm{data}}, \tilde{p}_{\mathrm{data}}) \geqslant 0$ is low (reconstruction error).

Guided by the theoretical analysis, we are motivated to seek such "qualified" manifold parametrizations for 3D graph DGMs, that is, $\overrightarrow{h}$ and $\overleftarrow{h}$ with low latent dimension $D''$ and low reconstruction error, which presents a non-trivial trade-off. The intertwined symmetries of 3D graphs present additional challenge to the latent space. Our solutions are cascaded 3D graph AEs on decomposed features and separate symmetries, as described in Sec. 3.1.

### 3.3 3D GRAPH GENERATION WITH EQUIVARIANT CONDITIONS

We are further interested in conditional generation of 3D graphs given another 3D object (e.g. a 3D graph $\mathcal{M}_{\mathrm{cond}}$), for instance, generating 3D molecules binding to given target-protein structures (Liu et al., 2022a; Guan et al., 2023a). Here the SE(3)-equivariant constraint needs to be satisfied, i.e., generated 3D graphs would translate/rotate accordingly w.r.t. conditional objects.

Since the latent space is constructed invariant to permutation and SE(3) transformations, latent 3D graph diffusion can be directly extended to conditional generation on equivariant attributes, featur-

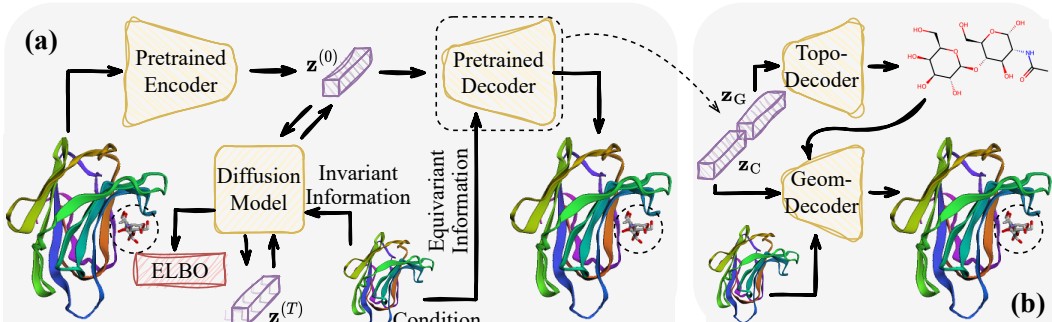

**Figure 3:** (a) Pipeline overview of latent 3D graph diffusion conditional on equivariant 3D objects. (b) Autoencoder architecture on the 3D graph topology and complex geometry.

ized as $\mathbf{x}_{\text{cond}}$, by simply overwriting the score estimator as $f_\theta(\mathbf{z}^{(t)}, \mathbf{x}_{\text{cond}}, t)$. We extend our pipeline accordingly into equivariant conditional generation (see Fig. 3(a) for overview).

**Complex encoding.** We keep the topological AE as in Sec. 3.1 and feed the geometrical AE with the 3D graphs of the complex (of generated molecules and conditional proteins) (Stärk et al., 2022).

**Equivariant decoding.** The latent space is still constructed invariant to (i.e. insensitive to) the rigid-body pose of conditional objects, i.e., the *absolute* geometry of conditions. Since conditional objects are always accessible during decoding/generation, we additionally feed them into the geometric decoder to provide such information as shown in Fig. 3(b), where the decoder is implemented to be equivariant based upon (Stärk et al., 2022). The auto-encoding processes are then formulated as:

$$\text{Encoding:} \quad \mathbf{z}_{\text{G}} = \overrightarrow{h}_{\text{enc,G}}(\mathcal{G}), \quad \mathbf{z}_{\text{C}} = \overrightarrow{h}_{\text{enc,C}}(\mathcal{C}, \mathcal{M}_{\text{cond}}), \quad \mathbf{z} = [\mathbf{z}_{\text{G}}; \mathbf{z}_{\text{C}}];$$

$$\text{Decoding:} \quad \bar{\mathcal{G}} = \overleftarrow{h}_{\text{dec,G}}(\mathbf{z}_{\text{G}}), \quad \bar{\mathcal{C}} = \overleftarrow{h}_{\text{dec,C}}(\bar{\mathcal{G}}, \mathcal{M}_{\text{cond}}, \mathbf{z}_{\text{C}}). \tag{4}$$

Thereafter, during latent diffusion, we learn to restore the latent distribution of topology and relative geometry given conditional objects as $p_\theta(\mathbf{z}|\mathcal{M}_{\text{cond}})$, which is achieved by overwriting the score estimator as $f_\theta(\mathbf{z}^{(t)}, \mathcal{M}_{\text{cond}}, t)$. See Append. C.1 for more details of architectures.

**Graph self-supervised learning as auxiliary pretraining objectives.** We also explore how 3D graph AEs constructed differently would impact latent diffusion performance. We leverage graph self-supervised learning (You et al., 2020a) to regularize the latent space, by acting as the auxiliary objective (i.e. multi-task learning) during topological and geometric AE pretraining. See Append. B.3 for more details. Inspired by (Kwon et al., 2022), we conjecture the better "semantics-aware" latent space is more amenable to DGMs capturing distributions. We thus quantify the extent of semantics-awareness and verify it is indeed indicative of 3D graph generation quality (Sec. 4.2).

## 4 EXPERIMENTS

We evaluate the proposed pipeline, (self-supervised) latent 3D graph diffusion, in three scenarios of real-world applications on 3D molecules: unconditional generation (Hoogeboom et al., 2022) (Sec. 4.1), invariant generation conditioned on quantum properties (Sec. 4.2), and equivariant generation conditioned on protein targets (Guan et al., 2023a) (Sec. 4.3). Before detailing the results, we briefly summarize our main findings as follows.

- Latent 3D graph diffusion generates topologically more valid and geometrically as stable molecules against competitors, for which AE's reconstruction quality is critical (results (i,vi)).
- Latent diffusion can better capture data distribution, even in the "out-of-distribution" scenario. Such capability is related to "semantic awareness" of the latent space (results (ii,iv,v,vii)).
- An additional bonus of latent diffusion is training efficiency: It is trained faster in the lower-dimensional latent space, by an order of magnitude than competitors (results (iii,vi)).

### 4.1 UNCONDITIONAL GENERATION OF 3D MOLECULES

**Configurations.** We pretrain our topological and geometric AEs on the large-scale public databases as ChEMBL (Gaulton et al., 2012) and PubChemQC (Nakata & Shimazaki, 2017), respectively, which can be repetitively utilized in almost all later experiments. We benchmark our method on the datasets of QM9 (with 100K smaller molecules up to 29 atoms) (Ramakrishnan et al., 2014) and

Drugs (with 450K larger molecules up to 181 atoms) (Axelrod & Gomez-Bombarelli, 2022) following (Hoogeboom et al., 2022; Wu et al., 2022; Vignac et al., 2023; Morehead & Cheng, 2023). The performance is evaluated on both validness of generated molecules (whether obeying stereochemical constraints) and distribution discrepancy with training data (whether capturing the distribution of observed molecules). See Append. D.1 for more details.

We compare with state-of-the-art (SOTA) competitors, including non-diffusion based models ENF (Satorras et al., 2021), G-Schnet (Gebauer et al., 2019) and diffusion models GDM, EDM (Hoogeboom et al., 2022; Wu et al., 2022), MiDi (Vignac et al., 2023), GCDM (Morehead & Cheng, 2023), GraphLDM, and GeoLDM (Xu et al., 2023).

**Results.** The results of unconditional generation are shown in Tabs. 2 and 3. We obtain the following observations based on the validness and distribution assessments.

**Table 2:** Unconditional generation evaluation on validness of 3D molecules. Valid: proportion of (POF) chemically valid molecules; Valid&Uni: POF chemically valid and unique molecules; AtomSta: POF atoms with correct valency; MolSta: POF molecules without unstable atoms. Numbers(std) in **red** are the best results.

| Methods | QM9 | | | | Drugs | | **Mean** |
|---|---|---|---|---|---|---|---|
| | Valid | Valid&Uni | AtomSta | MolSta | Valid | AtomSta | |
| ENF | 40.2 | 39.4 | 85.0 | 4.9 | – | – | 42.37 |
| G-Schnet | 85.5 | 80.3 | 95.7 | 68.1 | – | – | 82.40 |
| GDM | – | – | 97.0 | 63.2 | 90.8 | 75.0 | 81.50 |
| GDM-Aug | 90.4 | 89.5 | 97.6 | 71.6 | 91.8 | 77.7 | 86.43 |
| EDM | 91.9(0.5) | 90.7(0.6) | 98.7(0.1) | 82.0(0.4) | 92.6 | 81.3 | 89.53 |
| EDM-Bridge | 92.0 | 90.7 | 98.8(0.1) | 84.6(0.3) | 92.8 | 82.4(0.8) | 90.21 |
| GCDM | 94.8(0.2) | 93.3(0.0) | 98.7(0.0) | 85.7(0.4) | – | **89.0**(0.8) | 92.30 |
| MiDi | 97.9 | 97.0 | 97.9 | 84.0 | 78.0 | 82.2 | 89.50 |
| GraphLDM | 83.6 | 82.7 | **97.2** | 70.5 | 97.2 | 76.2 | 84.56 |
| GraphLDM-Aug | 90.5 | 89.5 | 97.9 | 78.7 | 98.0 | 79.6 | 89.03 |
| GeoLDM | 93.8(0.4) | 92.7(0.5) | **98.9**(0.1) | **89.4**(0.5) | 99.3 | 84.4 | 93.08 |
| Ours | **100.00**(0.00) | 95.27(0.25) | 97.57(0.02) | 86.87(0.23) | **100.00**(0.00) | 80.51(0.08) | **93.37** |

**Table 3:** Unconditional generation evaluation on distribution discrepancy with test data. NLL denotes negative log-likelihood values estimated by diffusion models, and other metrics represent total variation distances ($\times$1e-2) of certain molecular properties, between generated and observe molecules, the lower the better.

| Methods | NLL | MW | ALogP | PSA | QED | FCD | Energy |
|---|---|---|---|---|---|---|---|
| EDM | -1.22 | 2.89(0.38) | **0.85**(0.12) | 2.37(0.18) | **0.87**(0.05) | 58.04(0.39) | 2.81(0.29) |
| Ours | **-3.48** | **2.52**(0.39) | 0.91(0.10) | **1.22**(0.12) | 1.04(0.05) | **47.66**(3.42) | **1.87**(0.18) |

**(i) Quality of AEs is critical for latent diffusion validness performance.** Our latent diffusion generates more valid and diverse molecules in topology and as stable molecules in geometry versus SOTAs, as shown in Tab. 2. We realize these two perspectives of validness measurements are actually linked to the reconstruction quality of topological and geometric features of 3D graphs, that means, the quality of AEs is the bottleneck of validness assessment. As numerically reflected in Tab. 1, the topological AE is better trained for reconstruction than geometric in the sense of molecular data, which interprets the above validness results. This phenomenon also echoes our motivational analysis in Sec. 3.2 on AEs. Overall, our method leads to the best validness performance on average compared to SOTAs, credit to the well-constructed and trained 3D graph AE.

**(ii) Diffusion in latent space better captures data distributions.** Compared to EDM in the original 3D graph space, latent diffusion generates molecules of more similar property distributions as training data in 5 out of 7 metrics, as shown in Tab. 3. Results in other metrics such as Hellinger distance and Wasserstein metric are presented in Append. E.1. We also provide the probabilistic measure of negative log-likelihood (normalized on variables to remove the dimensionality bias) estimated by diffusion models, to demonstrate that our approach

**Table 4:** Training time per epoch (improvement) in unconditional generation. s: seconds; m: minutes.

| Methods | QM9 | Drugs |
|---|---|---|
| EDM | 146s | 349m |
| GeoLDM | 356s | 1307m |
| Ours | **12**s ($\downarrow$12$\times$) | **10**m ($\downarrow$34$\times$) |

appropriately captures data distributions. The results again echo our motivational analysis in Sec. 3.2 on latent diffusion. Another more intuitive interpretation is that the lower-dimensional latent embeddings preserve the more semantically important bits of data (Rombach et al., 2022), facilitating diffusion to learn distributions of molecules with similar properties.

Furthermore, since latent diffusion is trained in the (Euclidean) space of lower dimensionality and does not rely on the complicated message passing mechanism in score estimators, it provides an additional bonus of (**iii**) **12 / 34 times faster training** on the smaller / larger dataset as in Tab. 4.

## 4.2 Conditional Generation on (Invariant) Quantum Properties

**Configurations.** We benchmark our approach on QM9 (Ramakrishnan et al., 2014) following (Hoogeboom et al., 2022) where molecules are annotated with six quantum properties which are invariant to SE(3) transformations. Models take additional featurized properties as inputs, and are evaluated with the mean absolute error (MAE) between the conditional and oracle-predicted properties. We also examine whether models can correctly generate molecules conditional on properties different from training, which we refer as the "out-of-distribution" setting (OOD, versus ID).

We compare with the representative EDM (Hoogeboom et al., 2022), which however poses a strong bias between molecular sizes and property ranges. We argue it restricts the applicability of diffusion models, and thus remove the bias for a more realistic evaluation. See Append. D.2 for more details.

**Results.** The results of conditional generation on invariant attributes are shown in Tab. 5. We achieve the following observations through comparing between w/o and w/ latent diffusion in ID and OOD, and w/o and w/ applying graph self-supervised learning (GSSL) during AE training.

**Table 5:** Conditional generation on six quantum properties(unit) evaluation. Numbers represent the mean absolute error between conditional and oracle-predicted properties (Satorras et al., 2021), the lower the better. ID: in-distribution; OOD: out-of-distribution; GSSL: graph self-supervised learning.

| | Methods | $\alpha(\text{Bohr}^3)$ | $\Delta\varepsilon(\text{meV})$ | $\varepsilon_{\text{H}}(\text{meV})$ | $\varepsilon_{\text{L}}(\text{meV})$ | $\mu(\text{D})$ | $C_v(\frac{\text{cal}}{\text{mol}}\text{K})$ |
|---|---|---|---|---|---|---|---|
| | Random | 41.00 | 193.36 | 103.30 | 121.83 | 8.40 | **13.56** |
| ID | EDM | 20.15 | 287.00 | 158.70 | 166.20 | 7.01 | 13.63 |
| | Ours | **15.56** | **107.14** | **54.62** | **63.08** | **6.33** | 13.66 |
| | Ours-GSSL | 16.43 | 113.15 | 55.03 | 66.53 | 9.22 | 13.65 |
| | Random | 73.03 | **344.43** | 183.00 | 217.01 | **14.96** | **24.15** |
| OOD | EDM | 55.70 | 1561.9 | 1196.80 | 228.20 | 19.13 | 38.42 |
| | Ours | 32.06 | 363.13 | 109.30 | **178.69** | 22.18 | 31.12 |
| | Ours-GSSL | **30.07** | 388.31 | **103.86** | 179.41 | 26.89 | 40.82 |

(**iv**) **Latent diffusion leads to better generation quality conditional on ID or OOD properties.** Our latent diffusion achieves lower MAEs than EDM for 5 out of 6 ID and OOD properties. The greater challenges appear in the OOD setting (where the random baseline is pretty strong) and also the benefits of latent diffusion. This again demonstrates the compact semantics-focused latent space is more amenable for diffusion model learning, than the 3D graph space.

For a deeper understanding of what is learned by latent diffusion, we visualize the generated molecules conditional on different polarizability values ($\alpha$) in Append. E.2, which are expected less isometrically shaped for larger $\alpha$ (Wu et al., 2012). We find our model learns to generate molecules of larger $\alpha$ by increasing the chain length (Khan et al., 2002; Rodrigues et al., 2023). This behavior is different from that in EDM, which needs to prescribe the molecular size before generation. It points to a direction to explicitly bridging between latent and semantics, as in (Liu et al., 2022b).

(**v**) **Graph self-supervised learning (GSSL) improves OOD generation quality if further incorporating "semantics-awareness".** Applying GSSL during AE training improves 2 out of 6 metrics in the OOD setting. We additionlly explore on when GSSL would benefit latent diffusion. Based on the previous results and inspired by (Kwon et al., 2022), we conjecture the more "semantics-aware" latent space is easier for DGMs to capture distributions. We thus quantify such concept with the homogeneity ratio (Kwon et al., 2022) in the latent space, i.e., how consistently certain direction contributes to increasing property values for all data points. We find only for properties $\alpha$ and $\varepsilon_{\text{H}}$, GSSL improves the homogeneity ratio for certain direction as shown in Figs. 10 & 11, corresponding to the observed improvements of OOD results. See Append. E.2 for more details. This indicates a future potential to design semantics-specific GSSL tasks to boost latent diffusion, if prior knowledge is presented for the designated generation scenario.

## 4.3 Conditional Generation Binding to (Equivariant) Protein Targets

**Configurations.** We retain the topological AE, pretrain complex geometric AE and benchmark our method on the dataset of CrossDocked (100K complexes) (Francoeur et al., 2020) following

(Guan et al., 2023a). The evaluation is on the potentness of the generated 3D molecules justified by (in topology) drug-likeness (QED), synthesizability (SA), and (in geometry) binding affinity with protein targets computed with AutoDock Vina (Huey et al., 2012) (HiAff: the proportion of generated molecules with higher affinity than reference). See Append. D.3 for more details.

We compare our method with SOTA competitors including non-diffusion based models LiGAN (Ragoza et al., 2022), GraphBP (Liu et al., 2022a), AR (Luo et al., 2021), Pocket2Mol (Peng et al., 2022) and diffusion models TargetDiff (Guan et al., 2023a), DecompDiff (Guan et al., 2023b), DiffS-BDD (Schneuing et al., 2022). Since competitors are all pocket-dependent that pocket information is utilized during generation, for a fair comparison, we also use it via filtering the closest molecules to pocket residues after generation. The result of the pocket-free version without filtering is also reported for reference.

**Table 6:** Conditional generation on protein binding targets evaluation. Assessment metrics QED/SA & Vina scores are calculated with RDKit (Landrum, 2013) & AutoDock (Huey et al., 2012), respectively.

| Methods | Time↓ | QED↑ | SA↑ | HiAff↑ | Vina↓ | VDock↓ | Vina (Top-10%)↓ | Diversity↑ |
|---|---|---|---|---|---|---|---|---|
| LiGAN | – | 0.39 | 0.59 | 21.1% | – | -6.33 | – | 0.66 |
| GraphBP | – | 0.43 | 0.49 | 14.2% | – | -4.80 | -7.16 | 0.79 |
| AR | 211m | 0.51 | 0.63 | 37.9% | -5.75 | -6.75 | – | 0.70 |
| Pocket2Mol | 390m | 0.56 | **0.74** | 48.4% | -5.14 | -7.15 | -8.71 | 0.69 |
| TargetDiff | 340m | 0.48 | 0.58 | **58.1**% | -5.47 | -7.80 | -9.66 | 0.72 |
| DiffSBDD | – | 0.46 | 0.55 | – | **-7.33** | – | -9.92 | 0.75 |
| DecompDiff | – | 0.45 | 0.61 | 64.4% | -5.67 | **-8.39** | – | 0.68 |
| Ours | **6**m (↓35×) | **0.60** | 0.71 | 48.08% | -5.23 | -6.85 | **-12.34** | **0.80** |

(**vi**) **Latent diffusion generates more potent molecules for target proteins.** For conditional generation of 3D molecules equivariantly to conditional protein structures (Tab. 6), latent diffusion generates molecules of the highest drug-likeness and diversity and the second-highest synthesizability. This superior performance in topology generation is consistent with our previous experiments, and not sensitive to using protein pockets or not. As to geometry generation, although our method underperforms in the population evaluation of binding affinity, without AutoDock Vina docking (Vina) or with docking to change geometries (VDock), partly due to increasing diversity, it significantly outperforms others in the top 10% most potent binders, following the DiffSBDD evaluation. Furthermore, pre-training 3D graph AE with GSSL (Sec. 3.3), which regularizes the learning of the latent space, improves the Vina Dock score (Tab. 17).

Qualitatively, we also find (**vii**) **latent diffusion has the potential to model the pocket distribution**. We visualize the generated 3D molecules before pocket filtering in Fig. 12, where the circle area is a potential pocket that is not reflected in the reference molecule, and our method generates molecules appropriately covering both the reference and potential pockets. This is of significance in real-world drug discovery when exploring druggable protein pockets is pressing (Pérot et al., 2010).

## 5 CONCLUSIONS

This paper proposes an effective and efficient pipeline for 3D graph generation, dubbed latent 3D graph diffusion. It captures the 3D graph distribution by first encoding them into the latent space, training a diffusion model accordingly, and then decoding back to 3D graphs. The key questions explored in this paper are: *why* latent diffusion is useful for 3D graphs (motivational analysis, Sec. 3.2), *how* to parametrize the 3D graph latent space (auto-encoding decomposed views, Sec. 3.1), and *what* latent spaces better benefit latent 3D graph diffusion (semantics-awareness, Sec. 4.2). In applications, we also extend latent diffusion to the more significant conditional generation on invariant attributes (Sec. 4.2) or equivariant 3D objects (Sec. 4.3). Experiments in drug discovery verify the superiority of our pipeline in both generation quality and diffusion training efficiency.

Although our study illuminates the potential of generative modeling in the latent space for complex non-Euclidean data, it remains intriguing how to explicitly regularize the latent space with specific priors beyond our GSSL strategies.

ACKNOWLEDGEMENT

This project was in part supported by US Army Research Office Young Investigator Award W911NF2010240 and the NSF AI Institute for Foundations of Machine Learning (Z. Wang) as well as the National Science Foundation (CCF-1943008 to Y. Shen). Portions of this research were conducted with the advanced computing resources provided by Texas A&M High Performance Research Computing.

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

## A PROOFS FOR ANALYSIS

### A.1 PROOF FOR PROPOSITION 1

**Proposition 1.** Assume DGM is trained to model $p_{\text{data}}(\mathbf{x}_{\text{M}})$ with $p_\theta(\mathbf{x}_{\text{M}})$, $\forall t \geq 0$ the score $\nabla_{\mathbf{x}} \ln p^{(t)}$ is $L$-Lipschitz, for some $\eta > 0$ the moment $\mathbb{E}_{p_{\text{data}}(\mathbf{x}_{\text{M}})} \|\mathbf{x}_{\text{M}}\|^{2+\eta}$ is finite, and $\forall t \geq 0$ the score estimation error is bounded that $\mathbb{E}_{p^{(t)}(\mathbf{x}_{\text{M}}^{(t)})} \|f_\theta(\mathbf{x}_{\text{M}}^{(t)}, t) - \nabla_{\mathbf{x}} \ln p^{(t)}(\mathbf{x}_{\text{M}}^{(t)})\|^2 \leq \varepsilon_{\text{score}}^2$. Denote the second moment $\mathsf{m} = \mathbb{E}_{p_{\text{data}}(\mathbf{x}_{\text{M}})} \|\mathbf{x}_{\text{M}}\|^2$ and suppose the DGM step size is 1. Then, it holds for the 3D graph DGM assessment:

$$\text{TV}(\tilde{p}_\theta, \tilde{p}_{\text{data}}) \lesssim \alpha(p_\theta, \Pi, \Omega)\Big(\sqrt{\text{KL}(p_{\text{data}}\|\mathcal{N}_{D'})}e^{-T} + (L\sqrt{D'} + L\mathsf{m} + \varepsilon_{\text{score}})\sqrt{T}\Big),$$

where $\alpha(\cdot)$ depends on both the score estimator architecture and the symmetric groups, and $\mathcal{N}_{D'}$ is the normal distribution of dimension $D'$ that $D' = N \times (D_{\text{A}} + 3)$.

*Proof.* We bound the distribution discrepancy of the total variation distance between generated and training 3D graph features with the following inequalities:

$$\text{TV}(\tilde{p}_\theta, \tilde{p}_{\text{data}}) \overset{\text{(a)}}{=} \alpha(p_\theta, \Pi, \Omega) \, \text{TV}(p_\theta, p_{\text{data}})$$

$$\overset{\text{(b)}}{\lesssim} \alpha(p_\theta, \Pi, \Omega) \left( \sqrt{\text{KL}(p_{\text{data}}\|\mathcal{N}_{D'})}e^{-T} + (L\sqrt{D'} + L\mathsf{m} + \varepsilon_{\text{score}})\sqrt{T} \right),$$

where (a) results from the definition of $\alpha(p_\theta, \Pi, \Omega) = \frac{\text{TV}(\tilde{p}_\theta, \tilde{p}_{\text{data}})}{\text{TV}(p_\theta, p_{\text{data}})}$, and (b) is the result from (Chen et al., 2022a) Theorem 2. Note by data processing inequality (Beaudry & Renner, 2011), $\alpha(p_\theta, \Pi, \Omega) \in [0, 1]$. We thus reach the conclusion.

**Symmetry-related term** $\alpha(p_\theta, \Pi, \Omega)$. How does the DGM architecture regard symmetry structures affects the generation performance of 3D graphs, as reflected in the term $\alpha(p_\theta, \Pi, \Omega)$. Specifically, if the DGM $p_\theta$ is constructed in/equivariant to permutation $\Pi$ and SE(3) transformations $\Omega$, e.g. in (Hoogeboom et al., 2022), $\alpha(p_\theta, \Pi, \Omega)$ reaches the minimal value. That is, $p_\theta$ satisfies:

$$p_\theta(\mathbf{x}_{\text{M}}) = \begin{cases} \tilde{p}_\theta([\mathbf{x}_{\text{M}}]_{\Pi,\Omega}), & \text{if} \quad \mathcal{I}(\mathbf{x}_{\text{M}}) = 1, \\ 0, & \text{otherwise}, \end{cases}$$

where $\mathcal{I}(\mathbf{x}_{\text{M}}) \in \{0, 1\}$ is the indicator function which specifies only one $\mathbf{x}_{\text{M}}$ in each $[\mathbf{x}_{\text{M}}]_{\Pi,\Omega}$ with value 1 and 0 for others. This means the DGM treats all members in one equivalent class the same and thus only learns to capture one data point in each class. The ultimate distribution can be recovered later via $p_\theta'(\mathbf{x}_{\text{M}}) = p_\theta(\mathbf{x}_{\text{M}}')p_\Pi p_\Omega$ where $\mathbf{x}_{\text{M}}' \in [\mathbf{x}_{\text{M}}]_{\Pi,\Omega}, \mathcal{I}(\mathbf{x}_{\text{M}}') = 1$, and $p_\Pi, p_\Omega$ denote the probability of the transformation operators which are uniformly sampled. Given an arbitrary learned distribution $q_\theta$ that retains $\tilde{q}_\theta = \tilde{p}_\theta$, we have:

$$\text{TV}(q_\theta, p_{\text{data}})$$

$$\overset{\text{(a)}}{=} \sup_{S \subset \mathbb{R}^{D'}} |\text{Pr}\{\mathbf{x}_{\text{M}} : \mathbf{x}_{\text{M}} \in S, \mathbf{x}_{\text{M}} \sim q_\theta\} - \text{Pr}\{\mathbf{x}_{\text{M}} : \mathbf{x}_{\text{M}} \in S, \mathbf{x}_{\text{M}} \sim p_{\text{data}}\}|$$

$$\overset{\text{(b)}}{=} \sum_{\mathbf{x}_{\text{M}} \in S^+} q_\theta(\mathbf{x}_{\text{M}}) - \tilde{p}_{\text{data}}([\mathbf{x}_{\text{M}}]_{\Pi,\Omega})p_\Pi, p_\Omega$$

$$\overset{\text{(c)}}{=} \sum_{[\mathbf{x}_{\text{M}}']_{\Pi,\Omega}} \sum_{\mathbf{x}_{\text{M}} \in [\mathbf{x}_{\text{M}}']_{\Pi,\Omega} \cap S^+} \left( q_\theta(\mathbf{x}_{\text{M}}) - \tilde{p}_{\text{data}}([\mathbf{x}_{\text{M}}']_{\Pi,\Omega})p_\Pi, p_\Omega \right)$$

$$\overset{\text{(d)}}{\leq} \sum_{[\mathbf{x}_{\text{M}}]_{\Pi,\Omega}, [\mathbf{x}_{\text{M}}]_{\Pi,\Omega} \cap S^+ \neq \emptyset} \tilde{q}_\theta([\mathbf{x}_{\text{M}}]_{\Pi,\Omega}) - \tilde{p}_{\text{data}}([\mathbf{x}_{\text{M}}]_{\Pi,\Omega})p_\Pi, p_\Omega$$

$$\overset{\text{(e)}}{=} \sum_{[\mathbf{x}_{\text{M}}']_{\Pi,\Omega}, [\mathbf{x}_{\text{M}}']_{\Pi,\Omega} \cap S^+ \neq \emptyset} \sum_{\mathbf{x}_{\text{M}} \in [\mathbf{x}_{\text{M}}']_{\Pi,\Omega}, \mathcal{I}(\mathbf{x}_{\text{M}})=1} \left( p_\theta(\mathbf{x}_{\text{M}}) - \tilde{p}_{\text{data}}([\mathbf{x}_{\text{M}}']_{\Pi,\Omega})p_\Pi, p_\Omega \right)$$

$$\overset{\text{(f)}}{\leq} \text{TV}(p_\theta, p_{\text{data}}),$$

where (a) results from the definition of the total variation distance, (b) results from the equivalent definition of the total variation distance in the $\ell_1$ distance, and denoting $S^+ = \{\mathbf{x}_{\text{M}} : q_\theta(\mathbf{x}_{\text{M}}) > \tilde{p}_{\text{data}}([\mathbf{x}_{\text{M}}]_{\Pi,\Omega})p_\Pi, p_\Omega\}$, (c) results from grouping the summation in the individual equivalent classes, (d) results from the inequality $q_\theta(\mathbf{x}_{\text{M}}) \leq \tilde{q}_\theta([\mathbf{x}_{\text{M}}]_{\Pi,\Omega})$, (e) results from the in/equivariant construction of $p_\theta$, and (f) results from the definition of the total variation distance.

Thus, given an arbitrary learned distribution $q_\theta$ without altering $\tilde{q}_\theta$, we always can reframe it to be in/equivariant as $p_\theta$ that $\alpha(p_\theta, \Pi, \Omega) \leq \alpha(q_\theta, \Pi, \Omega)$, leading to a tighter bound in Prop. 1.

## A.2 PROOF FOR PROPOSITION 2

**Proposition 2.** Assume there existing mappings $\overrightarrow{h} : \mathbb{R}^{D'} \to \mathbb{R}^{D''}, \overleftarrow{h} : \mathbb{R}^{D''} \to \mathbb{R}^{D'}$ that $D'' < D'$ and $\overleftarrow{h}$ is injective. Assume DGM now is trained in $\mathbb{R}^{D''}$ to model $\overrightarrow{p}_{\text{data}}(\mathbf{z}) = \Pr\{\mathbf{x}_{\text{M}} : \overrightarrow{h}(\mathbf{x}_{\text{M}}) = \mathbf{z}, \mathbf{x}_{\text{M}} \sim p_{\text{data}}\}$ with $p_\theta(\mathbf{z})$, and it is evaluated in $\mathbb{R}^{D'}$ on $\overleftarrow{p}_\theta([\mathbf{x}_{\text{M}}]_{\Pi,\Omega}) = \Pr\{\mathbf{z} : \overleftarrow{h}(\mathbf{z}) \in [\mathbf{x}_{\text{M}}]_{\Pi,\Omega}, \mathbf{z} \sim p_\theta\}$ (as in Propos. 1), and the assumptions in Propos. 1 retain for the score estimator $f_\theta$ and mapping distribution. Then, it holds:

$$\text{TV}(\overleftarrow{p}_\theta, \tilde{p}_{\text{data}}) \lesssim \text{TV}(\overleftrightarrow{p}_{\text{data}}, \tilde{p}_{\text{data}}) +$$
$$\bar{\alpha}(p_\theta, \overrightarrow{h}, \overleftarrow{h}, \Pi, \Omega)\Big(\sqrt{\text{KL}(\overrightarrow{p}_{\text{data}} \| \mathcal{N}_{D''})}e^{-T} + (L\sqrt{D''} + L\mathsf{m} + \varepsilon_{\text{score}})\sqrt{T}\Big),$$

where $\overleftrightarrow{p}_{\text{data}}([\mathbf{x}_{\text{M}}]_{\Pi,\Omega}) = \Pr\{\mathbf{x}'_{\text{M}} : \overleftarrow{h}(\overrightarrow{h}(\mathbf{x}'_{\text{M}})) \in [\mathbf{x}_{\text{M}}]_{\Pi,\Omega}, \mathbf{x}'_{\text{M}} \sim p_{\text{data}}\}$, and $\bar{\alpha}(\cdot)$ depends on both the latent diffusion architecture that $\bar{\alpha}(p_\theta, \overrightarrow{h}, \overleftarrow{h}, \Pi, \Omega) = \alpha(\overleftrightarrow{p}_\theta, \Pi, \Omega)$ if $\overleftrightarrow{p}_{\text{data}} = p_{\text{data}}$.

*Proof.* We bound the distribution discrepancy of total variation distance between generated (from submanifold then reconstructed) and training 3D graph features with the following inequalities:

$$\text{TV}(\overleftarrow{p}_\theta, \tilde{p}_{\text{data}})$$

$$\overset{(a)}{\leq} \text{TV}(\overleftrightarrow{p}_{\text{data}}, \tilde{p}_{\text{data}}) + \text{TV}(\overleftarrow{p}_\theta, \overleftrightarrow{p}_{\text{data}})$$

$$= \text{TV}(\overleftrightarrow{p}_{\text{data}}, \tilde{p}_{\text{data}}) + \frac{\text{TV}(\overleftarrow{p}_\theta, \overleftrightarrow{p}_{\text{data}})}{\text{TV}(p_\theta, \overrightarrow{p}_{\text{data}})}\, \text{TV}(p_\theta, \overrightarrow{p}_{\text{data}})$$

$$\overset{(b)}{=} \text{TV}(\overleftrightarrow{p}_{\text{data}}, \tilde{p}_{\text{data}}) + \bar{\alpha}(p_\theta, \overrightarrow{h}, \overleftarrow{h}, \Pi, \Omega)\, \text{TV}(p_\theta, \overrightarrow{p}_{\text{data}})$$

$$\overset{(a)}{\lesssim} \text{TV}(\overleftrightarrow{p}_{\text{data}}, \tilde{p}_{\text{data}}) + \bar{\alpha}(p_\theta, \overrightarrow{h}, \overleftarrow{h}, \Pi, \Omega)\Big(\sqrt{\text{KL}(\overrightarrow{p}_{\text{data}} \| \mathcal{N}_{D''})}e^{-T} + (L\sqrt{D''} + L\mathsf{m} + \varepsilon_{\text{score}})\sqrt{T}\Big),$$

where (a) is achieved by constructing an intermediate distribution and applying the triangle equality, (b) comes from the rewriting by denoting $\bar{\alpha}(p_\theta, \overrightarrow{h}, \overleftarrow{h}, \Pi, \Omega) = \frac{\text{TV}(\overleftarrow{p}_\theta, \overleftrightarrow{p}_{\text{data}})}{\text{TV}(p_\theta, \overrightarrow{p}_{\text{data}})}$ that we have $0 < \bar{\alpha}(p_\theta, \overrightarrow{h}, \overleftarrow{h}, \Pi, \Omega) \leq 1$ due to the data processing inequality, and (c) is the result from (Chen et al., 2022a) Theorem 2. We thus reach the conclusion.

Besides, when the latent mappings are capable of recovering the complete data distribution that $\overleftrightarrow{p}_{\text{data}} = p_{\text{data}}$, we have $\bar{\alpha}(p_\theta, \overrightarrow{h}, \overleftarrow{h}, \Pi, \Omega) = \frac{\text{TV}(\overleftarrow{p}_\theta, \overleftrightarrow{p}_{\text{data}})}{\text{TV}(p_\theta, \overrightarrow{p}_{\text{data}})} \overset{(a)}{=} \frac{\text{TV}(\overleftarrow{p}_\theta, \tilde{p}_{\text{data}})}{\text{TV}(\overleftarrow{p}_\theta, p_{\text{data}})} = \alpha(\overleftarrow{p}_\theta, \Pi, \Omega)$ where (a) results from the ideal reconstruction of $\overrightarrow{h}, \overleftarrow{h}$, injectivity of $\overleftarrow{h}$ and data processing inequality, which matches the symmetry-related term in Prop. 1.

# B ADDITIONAL DETAILS FOR 3D GRAPH AES

## B.1 PRELIMINARY EFFORTS IN BUILDING 3D GRAPH ONE-SHOT AES

Our preliminary efforts attempt to build a 3D graph AE simply following the topological graph AE workflow (Simonovsky & Komodakis, 2018), as shown in Fig. 4. It compromises the following components.

**Encoders & decoders.** Topology and geometry information is encoded in the latent embedding, through separate encoding processes and then concatenation. The topological encoder is built with graph attention networks (Veličković et al., 2017) and geometric with polarizable atom interaction neural networks (Schütt et al., 2021). The decoder takes the concatenated latent embedding as input to reconstruct topological and geometric features, built with transformer architectures (Vaswani et al., 2017) followed by three multi-layer perceptrons for nodes, edges, and coordinates, respectively.

**Permutation- and SE(3)-invariant loss.** We enforce permutation and SE(3) invariance in the optimization objective. Specifically, before calculating the mismatch measurement between the input

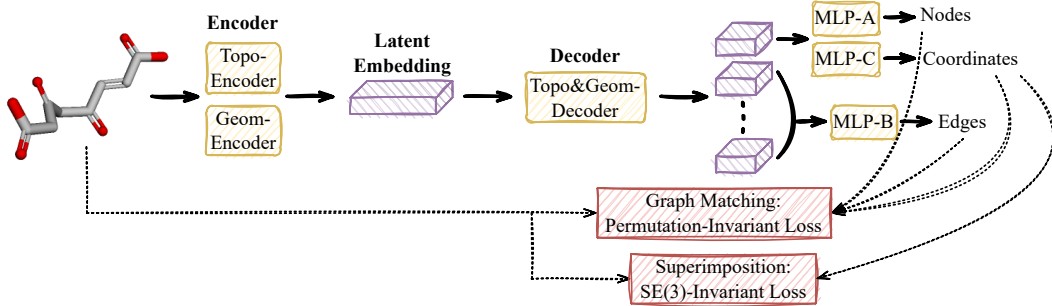

**Figure 4:** Pipeline overview of preliminary one-shot 3D graph AE.

and reconstructed 3D graph, we execute the neural graph matching algorithm (Yu et al., 2020) to sort out the correct node-to-node mapping, which leads to the permutation-invariant loss. Afterward, the superimposition algorithm (Kabsch, 1976) is executed to align the input and reconstructed coordinates with the global rotation and translation operations, which leads to the SE(3)-invariant loss. The final optimization objective then is the weighted sum of these losses.

## B.2 ADDITIONAL DETAILS FOR 3D GRAPH CASCADED AES

**Related works: Latent diffusion models.** Considerable research (Dai & Wipf, 2019; Yu et al., 2020) has focused on enhancing generative modeling capacity by exploring expressive generative models over the latent space. VQ-VAEs (Ragoza et al., 2022) proposed discretizing latent variables and leveraging autoregressive models to establish an expressive prior. (Ma et al., 2019), on the other hand, employed flow-based models as the latent prior, particularly for non-autoregressive text generation. Another line of research was inspired by the limitations of simple Gaussian priors in variational autoencoders (VAEs), which could not accurately match encoding posteriors and resulted in poor sample generation. To address this issue, (Dai & Wipf, 2019; Aneja et al., 2021) proposed using VAEs and energy-based models, respectively, to learn the latent distribution. More recently, various works have successfully developed latent diffusion models with promising results across different applications, such as image generation (Vahdat et al., 2021) and point clouds (Zeng et al., 2022). Among these methods, stable diffusion Models (Rombach et al., 2022) stand out for their impressive success in text-guided image generation, demonstrating remarkably realistic results.

**Encoders & decoders.** The topological AE is built with (Jin et al., 2020). Specifically, the topological encoder is composed of hierarchical layers of node message passing, attachment message passing, and motif message passing, and the decoder is composed of motif decoding, attachment decoding, and graph decoding. The geometric AE is built with (Satorras et al., 2021; Zhu et al., 2022a; Halgren, 1999). Specifically, the geometric encoder is composed of E(n) equivariant graph neural networks, and the decoder is composed of local node/edge message passing, global representation message passing, coordinate decoding and refinement Landrum (2013). We train both topological and geometric AEs for 2 days.

The topological encoder (Jin et al., 2020) consists of three Message Passing Networks (MPNs) to encode the hierarchical graph's three layers. Specifically, Atom Layer MPN encodes the atom layer of the hierarchical graph (HG). It takes embedding vectors of atoms and bonds as inputs. The network propagates message vectors between atoms over several iterations, culminating in the output of atom representation for each atom. In the topological decoder: (i) Decoder Function: The graph decoder incrementally expands the hierarchical graph to generate a molecule. It uses the same hierarchical MPN architecture to encode motifs and atoms in the partial hierarchical graph. (ii) Motif and Atom Vectors: At each generation step, the decoder produces motif vectors and atom vectors for the existing motifs and atoms in the graph. (iii) Motif Prediction: The model predicts the next motif to be attached, formulated as a classification task over the motif vocabulary. (iv) Attachment Prediction: The model predicts the attachment configuration of the next motif, focusing on the intersection of the motif with its neighbor motifs. This too is a classification task over an attachment vocabulary.

The geometric AE (Zhu et al., 2022a) consists of the following components. (i) Bond Representation Layer: Incorporates coordinate information into bond representations. The updated bond representation is derived from the previous bond representation, MLP operations on coordinate differences, and global representation. (ii) Atom Representation layer: Atom representations are updated using a Graph Attention Network version 2 (GATv2) to aggregate bond representations and then further refined with MLP operations and global representation. (iii) Global Molecule Representation: The global representation is updated by aggregating updated atom and bond representations with MLP operations. (iv) Conformation Construction: Each atom's conformation is predicted and normalized by centering the coordinates at the origin, ensuring numeric stability. The final prediction of the conformation is outputted by the last block with refinement Landrum (2013).

**Training via teacher forcing.** As depicted in Fig. 2(a), the decoding process is sequentially conducted, since the geometry decoding relies on the topology input. We remove such reliance during training to improve efficiency (that AEs can be trained in parallel and sampling is not needed during training), via applying teacher forcing: The topology input for the geometry decoding is the ground truth graph rather than generated, formulated as:

$$\text{Encoding:} \quad \mathbf{z}_G = \overrightarrow{h}_{\text{enc},G}(\mathcal{G}), \quad \mathbf{z}_C = \overrightarrow{h}_{\text{enc},C}(\mathcal{C}), \quad \mathbf{z} = [\mathbf{z}_G; \mathbf{z}_C];$$

$$\text{Decoding:} \quad \bar{\mathcal{G}} = \overleftarrow{h}_{\text{dec},G}(\mathbf{z}_G), \quad \bar{\mathcal{C}} = \overleftarrow{h}_{\text{dec},C}(\mathcal{G}, \mathbf{z}_C).$$

**Reconstruction quality of 3D graph AEs.** The results in Tab. 1 demonstrate the qualified reconstruction capability of 3D graph AEs. We further plot the topology and geometry reconstruction results for 3D graphs (3D molecules here) of varied properties, as shown in Figs. 5 and 6, respectively. Properties include number of heavy atoms (AtomNum), number of atoms (AtomNumWithHs), molecular weight (MW), octanol-water partition coefficient (ALOGP), number of hydrogen bond donors (HBD), number of hydrogen bond acceptors (HBA), polar surface area (PSA), number of rotatable bonds (ROTB), number of aromatic rings (AROM), and structural alerts (ALERTS). We do not see a strong correlation between certain properties and reconstruction quality.

We include in Append. E.1 more details on conceptual and numerical comparisons between oneshot AEs and our cascaded AEs.

### B.3 GRAPH SELF-SUPERVISED LEARNING IN AE TRAINING

**Related works: Graph self-supervised learning.** Graph self-supervised learning, surging recently, learns empirically more generalizable representations through exploiting vast unlabelled graph data (You et al., 2020a;b; 2021; 2022; You & Shen, 2022; Wei et al., 2022; Xu et al.) (please refer to (Xie et al., 2022) for a comprehensive review). The success of self-supervision hinges on big data and carefully designed pretext tasks, to enforce specific prior knowledge in graph models.

We apply graph self-supervised learning (GSSL) during AE training to regularize the 3D graph latent space. The adopted GSSL, graph contrastive learning (You et al., 2020a), is implemented as the auxiliary objective (i.e. multi-task learning), during which the node dropping augmentation is conducted on topological / geometric graphs, and the contrastive loss is optimized on the representations of augmented views.

## C 3D GRAPH GENERATION WITH EQUIVARIANT CONDITIONS

### C.1 ADDITIONAL DETAILS FOR 3D GRAPH AEs

**Encoders & decoders.** We retain the topological AE same as in Sec. 3.1. The geometric AE is built with (Guan et al., 2023a; Satorras et al., 2021; Stärk et al., 2022; Huey et al., 2012). Specifically, Specifically, the geometric encoder is composed of E(n) equivariant graph neural networks, and the decoder is composed of independent SE(3)-equivariant graph matching networks, multi-head SE(3)-equivariant attentions, docking transformation and refinement.

The conditional geometric AE (Guan et al., 2023a; Stärk et al., 2022) consists of the following components. (i) K-NN Graph Representations. Ligand Graph: Represented as a spatial k-nearest neighbor (k-NN) graph, using atoms as nodes. The graph includes atom pairs within a 4 Å distance cutoff. Receptor Graph: Constituted by residues as nodes, connected to the closest 10 other nodes

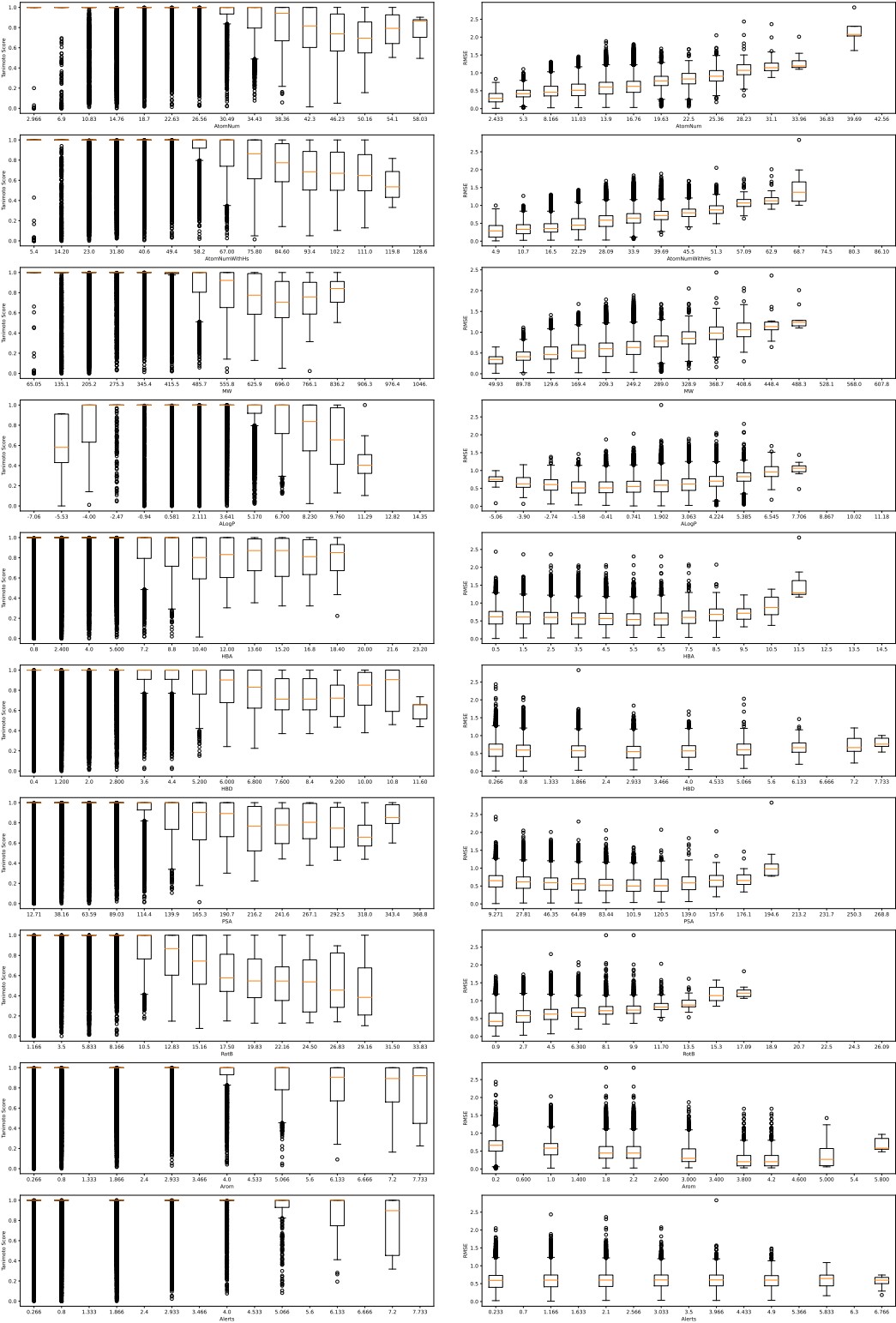

**Figure 5:** Topology reconstruction results versus different molecular properties.

**Figure 6:** Geometry reconstruction results versus different molecular properties.

within a 30 Å distance. (ii) IEGMN (Independent E(3)-Equivariant Graph Matching Network): This network combines Graph Matching Networks and E(3)-Equivariant Graph Neural Networks, allowing for joint transformation of features and 3D coordinates. (iii) Transformation Process: Ensures that any independent rotation and translation of input structures are precisely mirrored in the output, which is crucial for data-scarce problems like structural drug binding. (iv) Layer Architecture: Each layer involves specific operations that update the features and coordinates of the nodes, maintaining SE(3) invariance and employing shallow neural networks for various transformation functions.

# D  EXPERIMENTAL SETTINGS

## D.1  UNCONDITIONAL GENERATION

---

**Algorithm 1** Latent 3D Graph Diffusion Pipeline

---

**Input:** 3D graph data $\mathcal{D} = \{\mathcal{M}_1, ..., \mathcal{M}_N\} = \{(\mathcal{G}_1, \mathcal{C}_1), ..., (\mathcal{G}_N, \mathcal{C}_N)\}$, initial 2D encoder/decoder $\overrightarrow{h}_{\phi_{1,G}^{[0]}}, \overleftarrow{h}_{\phi_{2,G}^{[0]}}$, 3D encoder/decoder $\overrightarrow{h}_{\phi_{1,C}^{[0]}}, \overleftarrow{h}_{\phi_{2,C}^{[0]}}$, and diffusion model $p_{\theta^{[0]}}$.

▷ **Autoencoder training:**
**for** iteration $k = 1$ **to** $K$ **do**

   1. Latent encoding that $\mathcal{Z} = \{(\overrightarrow{h}_{\phi_{1,G}^{[k-1]}}(\mathcal{G}), \overrightarrow{h}_{\phi_{1,C}^{[k-1]}}(\mathcal{C})) : (\mathcal{G}, \mathcal{C}) \in \mathcal{D}\}$.

   2. Cascaded decoding that $\mathcal{D}' = \{(\overleftarrow{h}_{\phi_{2,G}^{[k-1]}}(\mathbf{z}_G), \overleftarrow{h}_{\phi_{2,C}^{[k-1]}}(\mathcal{G}, \mathbf{z}_C)) : (\mathbf{z}_G, \mathbf{z}_C) \in \mathcal{Z}\}$

   3. Updating parameters that $\phi_{i,U}^{[k]} = \phi_{i,U}^{[k-1]} - \eta \nabla_\phi \varepsilon_{\text{rec}}(\mathcal{D}, \mathcal{D}'), i \in \{1, 2\}, U \in \{G, C\}$.

**end for**

▷ **Diffusion model training:**

1. Latent encoding that $\mathcal{Z} = \{(\overrightarrow{h}_{\phi_{1,G}^{[K]}}(\mathcal{G}), \overrightarrow{h}_{\phi_{1,C}^{[K]}}(\mathcal{C})) : (\mathcal{G}, \mathcal{C}) \in \mathcal{D}\}$.

2. Updating parameters that **for** $k = 1$ **to** $K'$: $\theta^{[k]} = \theta^{[k-1]} - \eta \nabla_\theta \varepsilon_{\text{diff}}(\mathcal{Z}, p_{\theta^{[k-1]}})$.

▷ **Sampling:**

1. Sampling latent that $(\mathbf{z}_G, \mathbf{z}_C) \sim p_{\theta^{[K']}}$.

2. Reconstructing 3D graph $\mathcal{M} = (\mathcal{G}, \mathcal{C})$ that $\mathcal{G} = \overleftarrow{h}_{\phi_{2,G}^{[K]}}(\mathbf{z}_G), \mathcal{C} = \overleftarrow{h}_{\phi_{2,C}^{[K]}}(\mathcal{G}, \mathbf{z}_C)$.

---

**Background.** 3D molecule generation is a field within computational chemistry and drug discovery that focuses on the automated generation of three-dimensional structures of molecules. It plays a crucial role in understanding the properties and behavior of chemical compounds, as well as in the development of new drugs and materials. The process of generating 3D molecular structures computationally involves predicting the spatial arrangement of atoms and bonds in a molecule while satisfying various constraints, such as bond lengths, bond angles, and dihedral angles. 3D molecule generation has become an essential tool in various scientific and industrial applications. In drug discovery, it is employed to explore and optimize potential drug candidates, predict their binding to target proteins, and assess their pharmacokinetic properties. In materials science, 3D molecule generation aids in designing novel materials with specific properties, such as improved strength, flexibility, or conductivity.

**Evaluation (Hoogeboom et al., 2022).** Our model's performance is assessed by evaluating the chemical feasibility of the generated molecules to determine if the model effectively learns chemical rules from the data. The 'atom stability' metric measures the proportion of atoms in the generated molecules that have the correct valency, ensuring that the atom configurations are chemically valid. On the other hand, the 'molecule stability' metric represents the proportion of generated molecules in which all atoms maintain stable configurations. In essence, this metric assesses whether the entire molecular structure is chemically feasible. In addition to the stability metrics, we also report validity and uniqueness measurements. The 'validity' metric represents the percentage of generated molecules that are deemed valid according to RDKIT, a widely used software for molecular validation. This ensures that the generated molecules comply with well-established chemical rules and constraints. The 'uniqueness' metric, on the other hand, indicates the percentage of unique molecules among all the generated compounds, allowing us to quantify the model's ability to produce diverse and novel chemical structures.

The featurization of 3D molecules follows (Zhu et al., 2022b; Liu et al., 2021). The validness measurement is evaluated in the chemical feasibility of generated molecules, indicating whether the model can learn chemical rules from data (Xu et al., 2023), including topological validity and uniqueness, and geometric stability in atoms and molecules. The distribution measurement quantifies the discrepancy between the properties of observed and generated molecules, including molecular weight (MW), octanol-water partition coefficient (ALOGP), polar surface area (PSA), druglikeness (QED), Fréchet ChemNet distance (FCD) (Polykovskiy et al., 2020) and conformer energy. We also provide the probabilistic measure of negative log-likelihood (NLL) estimated by diffusion models, which is normalized on variables to remove the dimensionality bias (Hoogeboom et al., 2022).

## D.2 CONDITIONAL GENERATION ON (INVARIANT) QUANTUM PROPERTIES

**Evaluation.** In this task, our objective is to perform controllable molecule generation while adhering to desired properties. This capability proves valuable in practical applications such as material and drug design, where the focus is on discovering molecules that exhibit specific property preferences. To assess the performance, we use the QM9 dataset, which encompasses six properties: polarizability, orbital energies and their gap, Dipole moment, and heat capacity. To evaluate the model's ability to conduct property-conditioned generation, we follow the approach introduced by (Hoogeboom et al., 2022). We initially split the QM9 training set into two halves, each containing 50,000 samples. Next, we train a property prediction network on the first half of the dataset and subsequently train conditional models on the second half. During evaluation, we employ a range of property values, to conditionally draw samples from the generative models. We then use the property prediction network to calculate the corresponding predicted property values. To gauge the performance of property-conditioned generation, we report the Mean Absolute Error (MAE) between the actual property values and their predicted counterparts. A lower MAE indicates that the generated molecules closely align with the specified property conditions. Furthermore, to evaluate the bias of the property prediction network, we directly apply it to the second half of the QM9 dataset. A smaller MAE gap between the property-conditioned generation results and the QM9 dataset indicates a better property-conditioning performance, demonstrating the model's efficacy in generating molecules that closely match the desired property preferences.

The conditional properties are $\alpha(\text{Bohr}^3)$, tendency of a molecule to acquire an electric dipole moment when subjected to anexternal electric field; $\Delta\varepsilon(\text{meV})$, the energy difference between HOMO and LUMO; $\varepsilon_{\text{H}}(\text{meV})$, highest occupied molecular orbital energy; $\varepsilon_{\text{L}}(\text{meV})$, lowest unoccupied molecular orbital energy; $\mu(\text{D})$, dipole moment; $C_v(\frac{\text{cal}}{\text{mol}}\text{K})$, heat capacity at 298.15K.

We compare with the representative diffusion baseline (Hoogeboom et al., 2022). Different from (Hoogeboom et al., 2022) that poses overly strong inductive bias between the molecule size and properties during evaluation, i.e. $p(\mathcal{M}|\mathbf{x}_{\text{cond}}) \propto \sum_N p(\mathcal{M}_N|N, \mathbf{x}_{\text{cond}})p(N)\underline{p(\mathbf{x}_{\text{cond}}|N)}$, $\mathcal{M}_N$ is the molecule with $N$ atoms, where the range of $\mathbf{x}_{\text{cond}}$ given certain $N$ could be as narrow as 2 (versus random baseline 9.01 in (Hoogeboom et al., 2022)). We argue that such bias seriously restricts the applicability of the models. We thus conduct a more realistic evaluation for (Hoogeboom et al., 2022) on $p(\mathcal{M}|\mathbf{x}_{\text{cond}}) \propto \sum_N p(\mathcal{M}_N|N, \mathbf{x}_{\text{cond}})p(N)$ by removing the size correlation, and our model can implicitly learn the size distribution without prescribing it as prior $p(N)$ as in (Hoogeboom et al., 2022). We also construct the random baseline by randomly selecting molecules from the training set.

## D.3 CONDITIONAL GENERATION BINDING TO (EQUIVARIANT) PROTEIN TARGETS

**Evaluation.** The evaluation metrics are on the potentness of the generated 3D molecules justified by (in topology) drug-likeliness (QED), synthesizability (SA), and (in geometry) binding affinity with protein targets (HiAff: the proportion of generated molecules with higher affinity than the reference ligand). During the evaluation, we utilize pocket information by filtering the (top 33%) closest molecules to pocket residues after generation, where the pocket of the target protein is defined by the residues surrounding the reference ligand within 10Å (Liu et al., 2022a; Guan et al., 2023a). The Vina score is computed with AutoDock Vina, before docking (Vina) and after docking (VDock). Top-10% means only the lowest 10% of scores are evaluated.

---

**Algorithm 2** Conditional Latent 3D Graph Diffusion Pipeline

---

**Input:** Paired 3D graph data $\mathcal{D} = \{(\mathcal{M}_1, \mathcal{M}_{\mathrm{cond},1}), ..., (\mathcal{M}_N, \mathcal{M}_{\mathrm{cond},N})\} = \{((\mathcal{G}_1, \mathcal{C}_1), \mathcal{M}_{\mathrm{cond},1}), ..., ((\mathcal{G}_N, \mathcal{C}_N), \mathcal{M}_{\mathrm{cond},N})\}$, initial 2D encoder/decoder $\overrightarrow{h}_{\phi_{1,\mathrm{G}}^{[0]}}, \overleftarrow{h}_{\phi_{2,\mathrm{G}}^{[0]}}$, 3D encoder/decoder $\overrightarrow{h}_{\phi_{1,\mathrm{C}}^{[0]}}, \overleftarrow{h}_{\phi_{2,\mathrm{C}}^{[0]}}$, and diffusion model $p_{\theta^{[0]}}$.

▷ **Autoencoder training:**
**for** iteration $k = 1$ **to** $K$ **do**
    1. Latent encoding that $\mathcal{Z} = \{((\overrightarrow{h}_{\phi_{1,\mathrm{G}}^{[k-1]}}(\mathcal{G}), \overrightarrow{h}_{\phi_{1,\mathrm{C}}^{[k-1]}}(\mathcal{C}, \mathcal{M}_{\mathrm{cond}})), \mathcal{M}_{\mathrm{cond}}) : ((\mathcal{G}, \mathcal{C}), \mathcal{M}_{\mathrm{cond}}) \in \mathcal{D}\}$.
    2. Cascaded decoding that $\mathcal{D}' = \{((\overleftarrow{h}_{\phi_{2,\mathrm{G}}^{[k-1]}}(\mathbf{z}_{\mathrm{G}}), \overleftarrow{h}_{\phi_{2,\mathrm{C}}^{[k-1]}}(\mathcal{G}, \mathcal{M}_{\mathrm{cond}}, \mathbf{z}_{\mathrm{C}})), \mathcal{M}_{\mathrm{cond}}) : ((\mathbf{z}_{\mathrm{G}}, \mathbf{z}_{\mathrm{C}}), \mathcal{M}_{\mathrm{cond}}) \in \mathcal{Z}\}$
    3. Updating parameters that $\phi_{i,\mathrm{U}}^{[k]} = \phi_{i,\mathrm{U}}^{[k-1]} - \eta \nabla_\phi \varepsilon_{\mathrm{rec}}(\mathcal{D}, \mathcal{D}'), i \in \{1, 2\}, \mathrm{U} \in \{\mathrm{G}, \mathrm{C}\}$.
**end for**
▷ **Diffusion model training:**
1. Latent encoding that $\mathcal{Z} = \{((\overrightarrow{h}_{\phi_{1,\mathrm{G}}^{[K]}}(\mathcal{G}), \overrightarrow{h}_{\phi_{1,\mathrm{C}}^{[K]}}(\mathcal{C}, \mathcal{M}_{\mathrm{cond}})), \mathcal{M}_{\mathrm{cond}}) : ((\mathcal{G}, \mathcal{C}), \mathcal{M}_{\mathrm{cond}}) \in \mathcal{D}\}$.
2. Updating parameters that **for** $k = 1$ **to** $K'$: $\theta^{[k]} = \theta^{[k-1]} - \eta \nabla_\theta \varepsilon_{\mathrm{diff}}(\mathcal{Z}, p_{\theta^{[k-1]}})$.
▷ **Sampling:**
1. Sampling latent that $(\mathbf{z}_{\mathrm{G}}, \mathbf{z}_{\mathrm{C}}) | \mathcal{M}_{\mathrm{cond}} \sim p_{\theta^{[K']}}$.
2. Reconstructing 3D graph $\mathcal{M} = (\mathcal{G}, \mathcal{C})$ that $\mathcal{G} = \overleftarrow{h}_{\phi_{2,\mathrm{G}}^{[K]}}(\mathbf{z}_{\mathrm{G}}), \mathcal{C} = \overleftarrow{h}_{\phi_{2,\mathrm{C}}^{[K]}}(\mathcal{G}, \mathcal{M}_{\mathrm{cond}}, \mathbf{z}_{\mathrm{C}})$.

---

# E  MORE RESULTS AND DISCUSSIONS

## E.1  UNCONDITIONAL GENERATION

**Unconditional generation Evaluated in alternative metrics.** We also provide the results for unconditional generation evaluated in Hellinger distance and Wasserstein distance in Tabs. 7 & 8, respectively. Our improvement is consistent under various metrics.

**Table 7:** Unconditional generation evaluation on distribution discrepancy with training data. Metrics represent Hellinger distances ($\times$1e-2) of certain molecular properties, between generated and observe molecules.

| Methods | MW | ALogP | PSA | QED | Energy |
|---------|-----|-------|-----|-----|--------|
| EDM | 17.04(0.30) | 8.37(0.20) | 11.44(0.15) | 6.45(0.43) | 13.54(0.35) |
| Ours | 8.29(0.19) | 7.04(0.09) | 9.76(0.38) | 11.49(0.22) | 8.25(0.24) |

**Table 8:** Unconditional generation evaluation on distribution discrepancy with training data. Metrics represent Wasserstein distances ($\times$1e-2) of certain molecular properties, between generated and observe molecules.

| Methods | MW | ALogP | PSA | QED | Energy |
|---------|-----|-------|-----|-----|--------|
| EDM | 176.77(9.94) | 17.53(0.36) | 261.21(9.75) | 0.57(0.01) | 28.56(0.45) |
| Ours | 60.98(3.59) | 10.64(0.31) | 192.25(12.01) | 1.81(0.05) | 25.08(0.26) |

**Summary of differences across 3D-graph diffusion methods.** We summarize the conceptual difference across different methods in Tab. 9. We observe that:

- Lower LD latent space, better generation quality: Shifting from EDM and GeoLDM by reducing LD per node, better generation quality is achieved.

- Overly low LD latent space challenges its quality (in reconstruction) for generation: Shifting from GeoLDM to our one-shot model by further reducing the node factor in LD, it is difficult to achieve sufficient low reconstruction errors for for generation.

- More qualified latent space (in reconstruction), better generation quality: Shifting from our one-shot model to cascaded model, we achieve state-of-the-art generation results by building the 3D graph AE with low reconstruction errors.
- More "well-structured" latent space (in semantics), better OOD generation quality: Shifting from our cascaded model to self-supervised cascaded model, we achieve better OOD conditional generation results when the homogeneity metric is higher.

**Table 9:** Comparison across different methods, assuming the 3D graph has $N$ nodes, each of which is equipped with a $D$-dimensional invariant feature and 3-dimensional equivariant feature. Then the feature dimension of data is $N \times (D + 3)$, and we denote the latent dimension as $D'$.

| Methods | Topo/Geom Diffusion | Latent Dimension |
|---|---|---|
| EDM | Joint | $N \times (D + 3)$ |
| GeoLDM | Joint | $N \times (D' + 3)$ |
| Ours (One-Shot) | Joint | $D'$ |
| Ours (Cascaded) | Separate | $D'$ |

**Visualization of latent embeddings.** We provided t-SNE visualization of the latent embedding in Figs 7 & 8. We annotate data with four 2D and six 3D property values of molecules. Our visualization shows information of 2D properties is more preserved in topological embeddings, that data with similar properties tends to cluster. In addition, information of 3D properties is more preserved in geometric embeddings. The observation is also confirmed by quantitative evaluation, by computing the silhouette score of the latent embeddings w.r.t. properties.

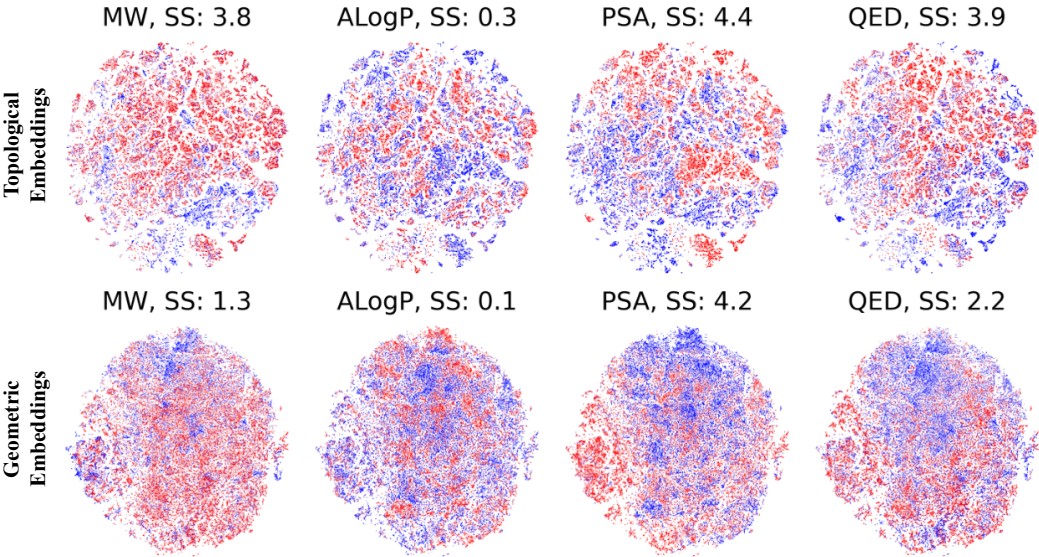

**Figure 7:** t-SNE visualization for topological and geometric latent embeddings. The colors are annotated by four 2D molecular properties: MW, ALogP, PSA, and QED. SS: Silhouette Score ($\times 100$).

**More ablation studies.** We here provide additional ablation studies. The experimental results of replacing the diffusion model with VAE are shown in Tabs. 10, 11 and 12. The experimental results of omitting the 3D latent space are shown in Tab. 13. The experimental results of training on less data (only 10% of the original data) are shown in Tabs. 14 & 15. And the experimental results of one-shot AE of different variants are shown in Tab. 16.

### E.2 CONDITIONAL GENERATION

**Visualization of conditionally generated molecules.** Please refer to Figs. 9, 12 for generated molecules conditional on different quantum properties and protein targets.

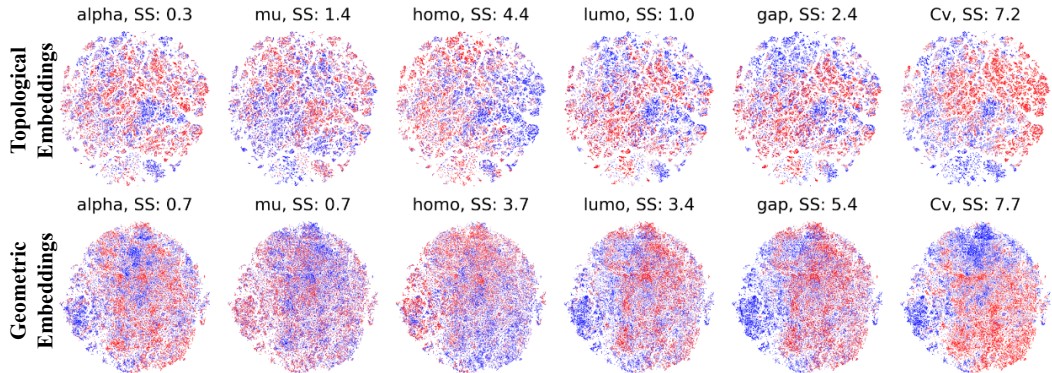

**Figure 8:** T-SNE visualization for topological and geometric latent embeddings. The colors are annotates by six 3D molecular properties. SS: Silhouette Score ($\times 100$).

**Table 10:** Unconditional generation evaluation on distribution discrepancy with training data. Metrics represent total variation distances ($\times$1e-2) of certain molecular properties, between generated and observe molecules, the lower the better. Reported are the mean values followed by the standard deviations in parentheses.

| Methods | MW | ALogP | PSA | QED | FCD | Energy |
|---|---|---|---|---|---|---|
| VAE | 64.30(22.78) | 66.29(23.57) | 64.25(23.37) | 66.60(23.57) | 2813.26(3.35) | 64.83(23.02) |
| Diffusion | 2.52(0.39) | 0.91(0.10) | 1.22(0.12) | 1.04(0.05) | 47.66(3.42) | 1.87(0.18) |

**Table 11:** Unconditional generation evaluation on distribution discrepancy with training data. Metrics represent Hellinger distances ($\times$1e-2) of certain molecular properties, between generated and observe molecules, the lower the better. Reported are the mean values followed by the standard deviations in parentheses.

| Methods | MW | ALogP | PSA | QED | Energy |
|---|---|---|---|---|---|
| VAE | 87.93(5.32) | 92.94(2.81) | 88.36(2.68) | 97.52(1.38) | 90.49(3.24) |
| Diffusion | 8.29(0.19) | 7.04(0.09) | 9.76(0.38) | 11.49(0.22) | 8.25(0.24) |

**Table 12:** Unconditional generation evaluation on distribution discrepancy with training data. Metrics represent Wasserstein distances ($\times$1e-2) of certain molecular properties, between generated and observe molecules, the lower the better. Reported are the mean values followed by the standard deviations in parentheses.

| Methods | MW | ALogP | PSA | QED | Energy |
|---|---|---|---|---|---|
| VAE | 1861.87(1944.16) | 114.75(47.24) | 1983.42(610.91) | 13.67(3.91) | 585.78(327.58) |
| Diffusion | 60.98(3.59) | 10.64(0.31) | 192.25(12.01) | 1.81(0.05) | 25.08(0.26) |

**Table 13:** Unconditional generation evaluation on distribution discrepancy with training data. Metrics represent total variation distances ($\times$1e-2) of certain molecular properties, between generated and observe molecules, the lower the better. Reported are the mean values followed by the standard deviations in parentheses.

| Methods | MW | ALogP | PSA | QED | FCD |
|---|---|---|---|---|---|
| 2D | 6.06(0.85) | 1.80(0.17) | 2.74(0.59) | 2.14(0.30) | 133.04(2.63) |
| 2D+3D | 2.52(0.39) | 0.91(0.10) | 1.22(0.12) | 1.04(0.05) | 47.66(3.42) |

**Rationale to enforce "semantic-awareness".** Several studies demonstrate if the latent space is aware of the specific semantic, there existing particular direction (for latent embeddings) to strength/weaken such semantic (Kwon et al., 2022; Liu et al., 2022b). We thus conjecture that such characteristics of the latent space could be linked to conditional generation performance: If a certain latent direction stands out to correlate with conditional property increasing/decreasing, diffusion models should easily capture it. We then leverage graph self-supervised learning (You et al., 2020a),

**Table 14:** Unconditional generation evaluation on distribution discrepancy with test data. Metrics represent total variation distances ($\times$1e-2) of certain molecular properties, between generated and observe molecules, the lower the better. Reported are the mean values followed by the standard deviations in parentheses.

| | Methods | MW | ALogP | PSA | QED | FCD | Energy |
|---|---|---|---|---|---|---|---|
| 100% Data | EDM | 2.89(0.38) | 0.85(0.12) | 2.37(0.18) | 0.87(0.05) | 58.04(0.39) | 2.81(0.29) |
| | Ours | 2.52(0.39) | 0.91(0.10) | 1.22(0.12) | 1.04(0.05) | 47.66(3.42) | 1.87(0.18) |
| | Ours-GSSL | 3.09(0.29) | 1.24(0.02) | 1.95(0.03) | 1.23(0.19) | 114.16(3.61) | 1.84(0.12) |
| 10% Data | EDM | 12.33(1.70) | 4.98(0.20) | 5.25(1.07) | 4.73(0.09) | 598.83(14.18) | 7.56(1.02) |
| | Ours | 11.63(1.90) | 2.72(1.23) | 4.06(1.02) | 3.25(0.44) | 401.30(31.87) | 6.80(1.06) |
| | Ours-GSSL | 10.04(2.08) | 3.74(0.20) | 3.65(0.40) | 2.24(0.24) | 387.80(10.86) | 5.53(0.65) |

**Table 15:** Conditional generation on polarizability evaluation. Numbers represent the mean absolute error between conditional and oracle-predicted properties (Satorras et al., 2021), the lower the better. Reported are the mean values followed by the standard deviations in parentheses. GSSL: graph self-supervised learning.

| Methods | 100% Data | 10% Data |
|---|---|---|
| Random | 41.00 | |
| EDM | 20.15 | 24.07 |
| Ours | 15.56 | 21.02 |
| Ours-GSSL | 16.43 | 20.49 |

**Table 16:** Reconstruction performance of AE workflows on molecule data, evaluated with topological accuracy (Recon.), and geometric root-mean-square error (RMSE).

| Methods | Recon.$\uparrow$ | RMSE $\downarrow$ |
|---|---|---|
| Random Init. | 0% | 1.86 |
| One-Shot AE | 0.80% | 1.80 |
| One-Shot AE w/ Higher Topo Weight | 0.30% | 1.66 |
| One-Shot AE w/ Higher Geom Weight | 0.60% | 1.60 |
| One-Shot AE w/ More Layers | 0.70% | 1.74 |
| Cascaded AE | 79.57% | 0.69 |

which is an effective way to regularize the latent space with specific priors. We adopt the standard node dropping prior. We further adopt the homogeneity ratio (Kwon et al., 2022) to quantify it. We surprisingly find the homogeneity ratio correlates with conditional generalization performance (Sec. 4.2 result (v)).

**Homogeneity ratio to quantify semantic-awareness.** We follow (Kwon et al., 2022) to calculate the homogeneity ratio to quantify semantic-awareness in the latent space. The idea is to check how the increasing/decreasing of the conditional properties is consistent toward a certain direction in the latent space. Specifically, given a latent embedding and its annotated property, (i) we search for its 50 nearest neighbors, (ii) calculate the angles between the center and neighbors and bin them into 10 groups, (iii) calculate the ratio (%) of property increasing within each group, and (iv) average the values on the dataset. If there existing outstanding ratios in a specific direction, it indicates the model can easily generate data toward the direction for the higher property, which we can it is semantic-aware; otherwise the latent space is not aware of the specific semantic and will reach 50% homogeneity ratios in all directions. The visualization of homogeneity ratios w.r.t. all six conditional properties are shown in Figs. 10 and 11.

**Ablation studies.** We include more ablation studies on conditional 3D molecule generation for given protein structures: Tab. 17 reported the impact of graph self-supervised learning (GSSL) and the resulting latent space regularization: topological evaluation of QED and topological plus geometric evaluation of Vina were actually worse, although VDock (after Vina docking thus changing geometries) improved. We expect that improving the cascaded AEs, especially the geometry AE, and testing the designs on new protein targets would better manifest the benefits of GSSL. Tab. 18 reported the impact of filtering generated molecules based on protein pockets (if known): high-

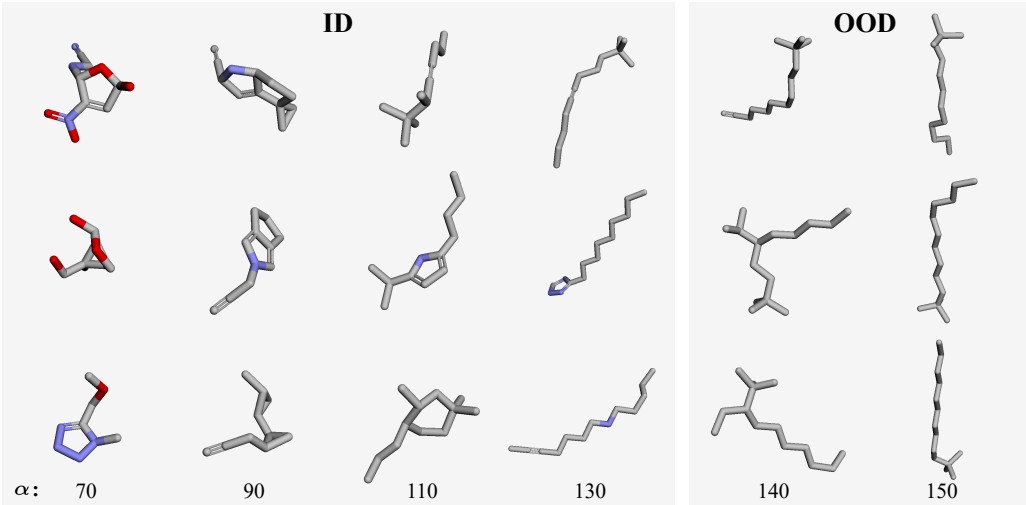

**Figure 9:** Visualization of molecules generated by latent 3D graph diffusion, conditional on different polarizability values ($\alpha$). Molecules of higher $\alpha$ are expected less isometrically shaped.

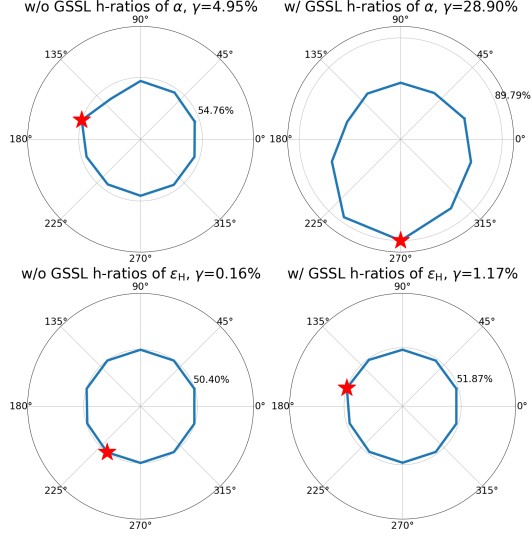

**Figure 10:** Polar plot for homogeneity ratios of $\alpha$ and $\varepsilon_H$, w/o and w/ GSSL. $\gamma$ is calculated as the absolute difference between maximum and median ratios.

affinity designs increased from 42% to 48% whereas topological evaluations such as QED and SA did not see significant changes. However, when no binding site information is available, our default latent diffusion model could uncover binding pockets (potentially novel), as observed in Fig. 12.

**Table 17:** Conditional generation on protein binding targets evaluation with graph self-supervised learning (GSSL). Numbers of QED/SA & Vina scores are calculated with RDKit (Landrum, 2013) & AutoDock (Huey et al., 2012), respectively.

| Methods | QED↑ | SA↑ | Vina↓ | VDock↓ | Vina (Top-10%)↓ |
|---|---|---|---|---|---|
| Ours | 0.60 | 0.71 | -5.23 | -6.85 | -12.34 |
| Ours-GSSL | 0.46 | 0.73 | -4.00 | -8.26 | -11.84 |

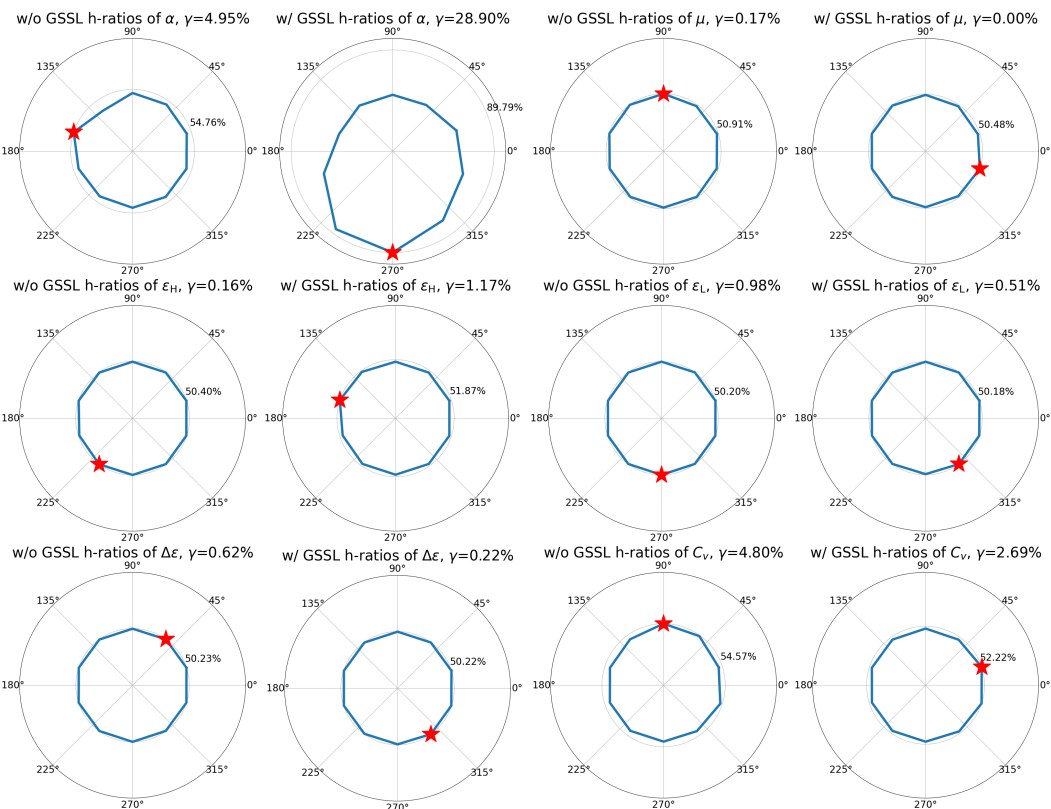

**Figure 11:** Polar plot for homogeneity ratios of $\alpha$, $\Delta\varepsilon$, $\varepsilon_H$, $\varepsilon_L$, $\mu$, $C_v$, w/o and w/ GSSL. $\gamma$ is calculated as the absolute difference between maximum and median ratios.

**Table 18:** Conditional generation on protein binding targets evaluation. Numbers of QED/SA & HiAff are calculated with RDKit (Landrum, 2013) & AutoDock (Huey et al., 2012), respectively. Pocket Unknown: latent diffusion without knowing the binding pocket and 3D molecules are generated for the conditional protein structures, which applies when no pocket information is known or novel pockets are desired (as discovered and illustrated in Fig. 12). Pocket Known: 3D molecules are still generated for the conditional protein structures but filtered based on the known binding pockets of protein structures.

| Methods | QED↑ | SA↑ | HiAff↑ |
|---|---|---|---|
| Pocket Known (Filtering molecules) | 0.60 | 0.71 | 48.08% |
| Pocket Unknown (Default) | 0.61 | 0.72 | 42.65% |

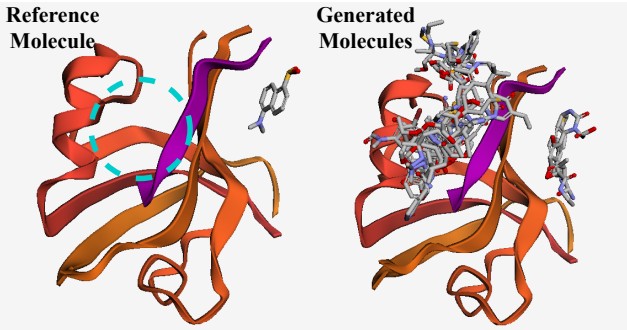

**Figure 12:** Visualization of the reference molecule and molecules generated by latent 3D graph diffusion, conditional on a protein binding target. The circled area is a potential binding pocket that is not reflected by the reference molecule.

