# OpenReview forum: "Latent 3D Graph Diffusion"
_ICLR.cc/2024/Conference — ICLR 2024 poster_

### Official Review · Reviewer_HVJQ · 2023-10-31

**Soundness:** 3 good
**Presentation:** 3 good
**Contribution:** 3 good
**Rating:** 6
**Confidence:** 3

**Summary:**

The paper "Latent 3D Graph Diffusion" explores using generative AI for 3D graph generation with symmetry-group equivariance. It emphasizes the importance of choosing the right latent space for diffusion, proposing a compact and symmetry-preserving space called "latent 3D graph diffusion." They extend this to conditional generation, showing its potential in molecular discovery with improved speed and quality compared to existing methods. The paper's contributions include motivational analysis, latent space construction, and extensions to conditional generation, all supported by experimental results. The conclusion suggests future research areas, including semantics-specific regularization of the latent space.

**Strengths:**

Strengths:
1.	Novel Approach: The paper addresses an important and under-investigated question regarding the latent space for 3D graph diffusion, introducing a novel concept of latent 3D graph diffusion. This approach is innovative and could lead to significant advancements in the field.

2.	Theoretical Foundation: The paper provides a theoretical analysis to motivate the use of latent spaces for 3D graph diffusion. The performance bound of 3D graph diffusion in a latent space is discussed, which adds a valuable theoretical perspective to the work.

3.	Comprehensive Contributions: The paper offers a multi-faceted approach to 3D graph generation, addressing the choice of latent space, the construction of compact and informative spaces, conditional generation, and regularization of the latent space. This comprehensive approach demonstrates the authors' commitment to advancing the field.

4.	Empirical Validation: The paper supports its claims with empirical results, showing that the proposed method outperforms existing techniques. Various settings are adopted, including unconditional generation, invariant generation conditioned on quantum properties, and equivariant generation conditioned on protein targets.

**Weaknesses:**

I have several questions:

1. What's the difference between a one-shot AE and a Cascaded AE? The paper claims that "a one-shot AE embeds and reconstructs molecule data, evaluating both structure topology and geometry simultaneously." How does a Cascaded AE differ? Are topology and geometry independent from each other in a Cascaded AE?

2. I would recommend adding an algorithm section to improve clarity. Based on my current understanding, a graph is initially encoded by an encoder in the latent space, then it undergoes diffusion in the latent space, and finally, it is transformed back to the graph space. Please correct me if I'm mistaken. Could you also clarify which diffusion model is used for the diffusion process in the latent space?

3. The paper asserts that "a latent space should possess (i) low dimensionality, (ii) low reconstruction error, and (iii) preserve group symmetry." How do you ensure that the autoencoder can meet these objectives? Have you conducted any analysis in this regard?

**Questions:**

See weaknesses.

---

> ### Author Response · Authors · 2023-11-21
> **Response to Reviewer HVJQ**
>
> **Q1.** What's the difference between a one-shot AE and a Cascaded AE? The paper claims that "a one-shot AE embeds and reconstructs molecule data, evaluating both structure topology and geometry simultaneously." How does a Cascaded AE differ? Are topology and geometry independent from each other in a Cascaded AE?
>
> **Answer.**
> - The reviewer understands correctly that the topological and geometric AEs encode and sequentially decode, independent from each other in the cascaded version, and in the one-shot version the two AEs jointly encode/decode. Please also refer to Appendices B.1 & B.2 for more information.
>
>
> **Q2.** I would recommend adding an algorithm section to improve clarity. Based on my current understanding, a graph is initially encoded by an encoder in the latent space, then it undergoes diffusion in the latent space, and finally, it is transformed back to the graph space. Please correct me if I'm mistaken. Could you also clarify which diffusion model is used for the diffusion process in the latent space?
>
> **Answer.**
> - The reviewer understands our pipeline correctly.
> - Following the suggestion, we add the detailed algorithm sections (Algorithms 1 & 2) to describe our latent 3D graph diffusion, beyond the overview Figures 2 & 3. They describe the processes of (conditional) in/equivariant latent encoding, latent diffusion, and sampling.
> - The adopted diffusion model is DDPM (https://arxiv.org/abs/2006.11239).
>
>
> **Q3.** The paper asserts that "a latent space should possess (i) low dimensionality, (ii) low reconstruction error, and (iii) preserve group symmetry." How do you ensure that the autoencoder can meet these objectives? Have you conducted any analysis in this regard?
>
> **Answer.**
> - The factors of (i) low dimensionality and (iii) preserving group symmetry are by design in the AE architecture, which is guaranteed.
> - The factor of (ii) low reconstruction error is achieved by building the 3D graph AE in a cascaded manner, as one of our contributions described in Section 3.1. The numerical comparison of different AEs is shown in Tables 1 & 11.

---

> > ### Comment · Reviewer_HVJQ · 2023-11-23
> >
> > Thanks for your rebuttal. Most of my concerns have been solved. I will keep my overall score.

---

> > > ### Author Response · Authors · 2023-11-23
> > > **Thank You**
> > >
> > > Thank you for the helpful comments and kind acknowledgment!

---

### Official Review · Reviewer_N36h · 2023-11-01

**Soundness:** 3 good
**Presentation:** 3 good
**Contribution:** 3 good
**Rating:** 6
**Confidence:** 5

**Summary:**

In this paper, the authors introduce the generation of 3D graphs characterized by symmetry group equivariances, a feature with useful implications for machine vision and molecular discovery.The paper focuses on the optimal latent space for 3D graph diffusion, emphasizing its benefits over traditional methods. By strategically using cascaded 2D-3D graph autoencoders, the authors reveal a model called "latent 3D graph diffusion". Notably, this innovative approach demonstrates remarkable efficacy when adapted to the molecular context and enhanced by graph self-supervised learning. The experimental results highlight its ability to rapidly generate remarkable 3D molecular conformations.

**Strengths:**

1. This study provides a detailed analysis of the underlying methodology. It illustrates the overall relationship between diffusion performance, latent space reconstruction quality, symmetry conservation, and data dimensionality. The depth of this analysis demonstrates the thoroughness and precision of the research.

2. In this work, researchers adopt graph contrastive learning as a strategy to refine and enhance latent space representations in 3D graph autoencoders (AEs). The approach stands out for its innovative nature.

3. In terms of application evaluation, the authors conduct an exhaustive evaluation of the model for various scenarios. This includes unconditional generation of 3D molecules, conditional generation based on (invariant) quantum properties, and conditional generation in relation to (equivariant) protein targets. These diverse evaluations reinforce the model's laudable versatility and adaptability.

**Weaknesses:**

1. For the objective of unconditional 3D molecular generation, I noticed the absence of results from "Ours-GSSL". Including these would be crucial to validate the effectiveness of GCL.

2. Regarding the unconditional 3D molecule generation, the 'molsta' metric, which gauges molecular integrity, seems paramount. From the data presented, your findings exhibit significant variances when juxtaposed against the previously introduced 3D latent diffusion model, GeoLDM. Additionally, the AtomSta and MolSta metrics for the Drug dataset appear to be omitted.

3. In the context of conditional generation based on (invariant) quantum properties, it's essential to evaluate both ID and OOD. Yet, the "Ours-GSSL" performance doesn’t exhibit a noteworthy distinction compared to "ours", "Random", and similar methods. In some instances, it even underperforms. This might cast a shadow on the robustness of conclusions articulated in Section 4.2, especially point (v).

4. Concerning the experiments related to conditional generation associated with (equivariant) protein targets, I'd recommend broadening the evaluation scope by incorporating metrics such as the Vina Score and Vina Dock in your findings. Furthermore, beyond Targetdiff, newer techniques like DiffSBDD[1] and DecompDiff[2] have emerged. Comparing your approach with these could provide a more holistic view. As per the metrics currently displayed, the method delineated in your manuscript doesn't seem to lead the pack.

Minor:

1. In Figure 4, there appear to be some omission errors; the circled references seem to be missing.

2. The structure of "Proofs for Analysis" in Appendix A could be refined for better readability.

[1] Schneuing, Arne, et al. "Structure-based drug design with equivariant diffusion models." arXiv preprint arXiv:2210.13695 (2022).

[2] Guan, Jiaqi, et al. "DecompDiff: Diffusion Models with Decomposed Priors for Structure-Based Drug Design." (2023).

**Questions:**

1. The design of latent diffusion appears to be complex. Have you considered simplifying it by merely applying KL divergence to $z^{{0}}$ and deriving a VAE-like model using the encoder and decoder as proposed in this paper? An ablation study should be added here.

2. Could there be a more comprehensive validation of GCL's efficacy in Section 4?

3. Given your assertion that your model "produces superior 3D molecules faster" owing to latent diffusion, could you perhaps contrast it with GeoLDM, a previously introduced 3D latent diffusion model?

4. Is it possible to conduct a more in-depth evaluation and comparative study of experiments related to the generation of conditions related to equivariant protein targets?

---

> ### Author Response · Authors · 2023-11-21
> **Response to Reviewer N36h (1)**
>
> **Q1.** For the objective of unconditional 3D molecular generation, I noticed the absence of results from "Ours-GSSL". Including these would be crucial to validate the effectiveness of GCL. Could there be a more comprehensive validation of GCL's efficacy in Section 4?
>
> In the context of conditional generation based on (invariant) quantum properties, it's essential to evaluate both ID and OOD. Yet, the "Ours-GSSL" performance doesn’t exhibit a noteworthy distinction compared to "ours", "Random", and similar methods. In some instances, it even underperforms. This might cast a shadow on the robustness of conclusions articulated in Section 4.2, especially point (v).
>
> **Answer.**
> - We provide additional results in Tables 15 & 16, on applying GSSL for unconditional generation and conditional generation with less training data.
> - Based on our experiments, we observed that the benefit of GSSL is mainly on:
>     - OOD scenarios when the GSSL prior is aligned with the condition semantics, as we stated in Section 4.2;
>     - Unconditional or ID scenarios when training data are insufficient. This is consistent with the role of GSSL in discriminative modeling (https://arxiv.org/abs/2010.13902).
> - For the OOD scenarios, though the improvement does not appear for all datasets, we know when it is improved: Specifically, we can measure the semantics-alignment by computing the homogeneity ratio as detailed in Appendix E.2. This indicates a future potential to design semantics-specific GSSL tasks for generative modeling.
>
> **Q2.** Regarding the unconditional 3D molecule generation, the 'molsta' metric, which gauges molecular integrity, seems paramount. From the data presented, your findings exhibit significant variances when juxtaposed against the previously introduced 3D latent diffusion model, GeoLDM. Additionally, the AtomSta and MolSta metrics for the Drug dataset appear to be omitted.
>
> **Answer.**
> - Our method is different from GeoLDM. We summarize the difference across different methods in Table 10, mainly lying in two perspectives: (i) latent dimension and (ii) the modeling of topological and geometric features.
>     - Latent dimension (LD). Assuming the feature dimension of data is $N \times (D+3)$. GeoLDM, diffusing at the latent space of nodes is of LD $N \times (D’+3)$. Guided by our theory in Section 3.2, we chase a lower dimensional latent space by compressing the node factor $N$, that our method at the other end of the spectrum of the minimum LD $D’$.
>     - Modeling of topological and geometric features. EDM, GeoLDM and our one-shot model jointly model the topological and geometric features for all $N$ nodes. However, our one-shot model cannot manage to reconstruct the feature per node for its overly low LD, that the node factor $N$ is compressed. Guided by our theory in Section 3.2, we develop the cascaded model, achieving satisfactory reconstruction performance while maintaining symmetry and sufficiently low LD.
> - The above aspects lead to different results: GeoLDM performs better in generating geometrically stable molecules, and our method performs better in generating topologically valid molecules. Both of them are important in real-world applications.
> - The evaluation is standardized as in EDM and GeoLDM, and we do report AtomSta for Drugs. Following EDM and GeoLDM, the MolSta metric is less informative to be evaluated in Drugs, since the ground-truth molecules in Drugs have 86.5% atom-level and nearly 0% molecule-level stability, due to which contain larger and more complex structures, creating errors during bond type prediction based on pair-wise atom types and distances.

---

> > ### Author Response · Authors · 2023-11-21
> > **Response to Reviewer N36h (2)**
> >
> > **Q3.** Concerning the experiments related to conditional generation associated with (equivariant) protein targets, I'd recommend broadening the evaluation scope by incorporating metrics such as the Vina Score and Vina Dock in your findings. Furthermore, beyond Targetdiff, newer techniques like DiffSBDD[1] and DecompDiff[2] have emerged. Comparing your approach with these could provide a more holistic view. As per the metrics currently displayed, the method delineated in your manuscript doesn't seem to lead the pack. Is it possible to conduct a more in-depth evaluation and comparative study of experiments related to the generation of conditions related to equivariant protein targets?
> >
> > **Answer.**
> > - We provide a more comprehensive evaluation including metrics of Vina Score, Vina Dock, and baselines of DiffSBDD and DecompDiff in Table 6.
> > - Overall, there is no single method that outperforms across all metrics. Our method is very competitive to achieve the best on 4 out of 8 metrics.
> > - Our method in general performs well in generating small molecules that are more drug-like, easier to synthesize, and more diverse compared to state-of-the-arts.
> > - Although our method underperforms in the population evaluation of Vina Score, Vina Dock, it significantly outperforms others on Vina Score (top 10%), following the DiffSBDD evaluation. This indicates that by modeling the distribution in the more compact and semantic-aware 3D graph latent space, it owns the potential to generate very potent small-molecule binders to the protein target. This would be very useful in real-world drug discovery to screen the effective drug candidates.
> >
> > **Q4.** In Figure 4, there appear to be some omission errors; the circled references seem to be missing. The structure of "Proofs for Analysis" in Appendix A could be refined for better readability.
> >
> > **Answer.**
> > - We thank the reviewers for comments. We have added the reference and will further refine the structure of Appendix A for better presentation.
> >
> > **Q5.** The design of latent diffusion appears to be complex. Have you considered simplifying it by merely applying KL divergence to  z0  and deriving a VAE-like model using the encoder and decoder as proposed in this paper? An ablation study should be added here.
> >
> > **Answer.**
> > - We provide the suggested ablation study in Tables 12-14.
> > - We observed that the diffusion model outperforms VAE here, and agreed with the findings in many fields, e.g. https://arxiv.org/abs/2011.13456.
> >
> > **Q6.** Given your assertion that your model "produces superior 3D molecules faster" owing to latent diffusion, could you perhaps contrast it with GeoLDM, a previously introduced 3D latent diffusion model?
> >
> > **Answer.**
> > - We thank the reviewer for the comment. We supplement the GeoLDM running time result in Table 4.

---

> ### Comment · Reviewer_N36h · 2023-12-01
>
> The author has addressed all my concerns. I will increase my score accordingly.

---

### Official Review · Reviewer_7uXM · 2023-11-01

**Soundness:** 3 good
**Presentation:** 3 good
**Contribution:** 2 fair
**Rating:** 6
**Confidence:** 4

**Summary:**

The paper proposes to perform 3D Graph (molecule) generation in latent space instead of during it directly in the 3D space as usually done now. The author show that diffusion in a lower dimensional space should be more efficient for diffusion. They also propose to use a cascaded auto encoder (AE) instead of a one-shot AE to build such latent space. These changes are shown to help improve the quality of generated samples and are also applicable in the conditioned case.

**Strengths:**

The authors provide a nice and sensible motivation, that smaller latent spaces should be more efficient for diffusion. The proposed approach also outperforms the chosen baselines. The proposed formulation is nicely extended to include conditioning on both scalar and 3D properties, which allows for a wider range of applications.

**Weaknesses:**

The novelty of the work is somewhat limited. It takes existing 3D molecule genration setup with diffusion models and transforms it into latent diffusion, which is a known concept in general and some latent 3D models do exist (e.g. GEOLDM), using mostly off the shelf architectures.

One of the main contributions of this work is the cascaded auto encoder and main motivation for why it is needed stems from the fact that one-shot AE effectively failed to train. I find it a bit strange that the one-shot AE fails to train completely as there are 3D molecule generative models that generate 2D and 3D graph features jointly (e.g. MIDI or the example I reference in the paragraph below). They do usually re-weight the losses for 2D and 3D terms, but they do work. So I would like to see a more detailed analysis on why it doesn't work and potentially an ablation on the one-shot AE architecture and losses. I can understand one-shot AE working worse than the proposed cascaded AE with teacher forcing, but it essentially performing like a random initialization makes me wonder if it was tested sufficiently carefully.

As I also point out below, it would make sense to use for example https://arxiv.org/pdf/2309.17296.pdf as the baseline instead of an older EDM model, even though that paper is somewhat recent, so I understand its exclusion for the initial version of the paper. Still it would be nice to have for the rebuttal.

**Questions:**

In section 3.1 it is stated that graph matching is used for the AE training loss to ensure permutation and SE3 invariance. This can be computationally expensive. How fast is the AE training? In terms of wall time, but also asymptotically (in O notation)? Without knowing this the comparison of training time of the proposed method vs EDM is also a bit complicated. As I understand Table 4. does not account for the AE training time (yes, it can be trained once, but it still needs training).

Also, since AE is trained on a much larger dataset, it might be fair to also pre-train the EDM on general molecule genration. It has been shown that pretraining helps molecule generation (https://arxiv.org/pdf/2309.17296.pdf). While this paper is quite recent it would still make more sense to compare against such properly tuned state-of-the-art molecular diffusion setup instead of vanilla EDM, which is an older baseline (first ever diffusion model for 3D molecule generation).

Why is MIDI, which is cited numerous times in the paper is not compared against in the experiments? It does in certain metrics perform a lot better than the EDM, which is used as the main baseline here.

Also, in 3.2 Setup sections authors say that connectivity is commonly later determined based on domain rules and cite MIDI among others. Its true, that such domain-based rules are used to recover connectivity in e.g. EDM, but in MIDI connectivity is modeled explicitly if I remember correctly?

---

> ### Author Response · Authors · 2023-11-21
> **Response to Reviewer 7uXM (1)**
>
> **Q1.** The novelty of the work is somewhat limited. It takes existing 3D molecule genration setup with diffusion models and transforms it into latent diffusion, which is a known concept in general and some latent 3D models do exist (e.g. GEOLDM), using mostly off the shelf architectures.
>
> **Answer.**
> - **Our work is not a direct extension of latent diffusion into 3D graphs.** We aim at generating 3D graphs of symmetry-group equivariance through diffusion. In other words, the quality of the generated graphs of topology and geometry is invariant / equivariant to a set of transformations (permutation group and Lie group E(3) of rotations and translations in our study).  This aim has deep roots in 3D molecular generation as demonstrated in our numerical experiments for desired quantum properties (invariant to E(3) perturbation of the 3D molecules) or desired protein binding (equivariant to E(3) perturbation of the 3D molecules); and it has broader application scopes for 3D graphs of symmetry-group equivariance in general.
> - **Contribution (i).** We provided theoretical analysis (Section 3.2) on what space should diffusion generative models operate on for generating 3D graphs of non-Euclidean structure and group-symmetry equivariance.  We addressed group-symmetry equivariance while deriving model performance bounds.  Specifically, Proposition 1 gave the upper bound of the diffusion model performances (as measured in total variation distance) operating in the original 3D graph domain ($D’$-dimensional); whereas Proposition 2 gave the upper bound in the latent space ($D’’$-dimensional). The latter led to a better upper bound of the diffusion model performances if reconstruction errors are low and the latent dimension is low, which motivates our approach of latent diffusion.
> - **Contribution (ii).** In pursuing the latent space as guided by our theoretical, motivational analyses, we constructed a 3D graph autoencoder of the following properties: (1) permutation and E(3) invariant and (2) sufficiently low reconstruction errors despite low latent dimension (these two present nontrivial trade-offs), which is under-explored.  We accomplished this by combining 2D graph autoencoder (for topology features) and point cloud autoencoder (for geometry features given topologies) in a cascade manner (Section 3.1).
> - **Contribution (iii).** To improve the out-of-domain (OOD) generation, we further incorporate graph self-supervised learning into learning the 3D graph autoencoder and the resulting latent space for diffusion (Section 3.1).  Our self-supervised latent diffusion has been largely unexplored for 3D graphs of group-symmetry equivariance.  Our numerical experiments demonstrated its effectiveness in improving OOD performances, which is a critical barrier for designing molecules with much improved properties or new properties.
> - **Contribution (iv).** We further extend our work into conditional generation of 3D graphs, which stems from practical applications and presents a new challenge.  Namely the new challenge is to maintain conditional equivariance, that is, the generated 3D graphs should be E(3)-invariant to the property condition and E(3)-equivariant to the protein binder’s structure condition. We addressed this challenge by introducing conditional equivariant geometric autoencoder in our latent diffusion pipeline (Section 3.3).
> - **Difference from GeoLDM.** We summarize the difference across different methods in Table 10, mainly lying in two perspectives: (i) latent dimension and (ii) the modeling of topological and geometric features.
>     - Latent dimension (LD). Assuming the feature dimension of data is $N \times (D+3)$. At  one end of the spectrum, EDM diffusing in the original space is of the maximum LD $N \times (D+3)$. The followup work, GeoLDM, diffusing in the latent space of nodes is of LD $N \times (D’+3)$. Guided by our theory in Section 3.2, we chase an even lower dimensional latent space by compressing the node factor $N$, that our method at the other end of the spectrum of the minimum LD $D’$.
>     - Modeling of topological and geometric features. EDM, GeoLDM and our implemented one-shot model jointly model the topological and geometric features for all $N$ nodes. However, our one-shot model cannot manage to reconstruct the feature per node for its overly low latent dimension, that the node factor $N$ is compressed. Guided by our theory in Section 3.2, we develop the cascaded model, achieving satisfactory reconstruction performance while maintaining symmetry and sufficiently low LD.

---

> > ### Author Response · Authors · 2023-11-21
> > **Response to Reviewer 7uXM (2)**
> >
> > **Q2.** One of the main contributions of this work is the cascaded auto encoder and main motivation for why it is needed stems from the fact that one-shot AE effectively failed to train. I find it a bit strange that the one-shot AE fails to train completely as there are 3D molecule generative models that generate 2D and 3D graph features jointly (e.g. MIDI or the example I reference in the paragraph below). They do usually re-weight the losses for 2D and 3D terms, but they do work. So I would like to see a more detailed analysis on why it doesn't work and potentially an ablation on the one-shot AE architecture and losses. I can understand one-shot AE working worse than the proposed cascaded AE with teacher forcing, but it essentially performing like a random initialization makes me wonder if it was tested sufficiently carefully.
> >
> > **Answer.**
> > - We provide the re-weighting results in Table 11.
> > - We tested one-shot AE extensively. We have conducted very extensive experiments on tuning the one-shot AE, including loss re-weighting, architecture variation, and hyperparameter optimization. The training does not converge and the reconstruction looks random, which turned us to the cascaded manner.
> > - The potential reason might lie in the dimensionality of our latent space, as described in the Q1 answer, that our latent space dimensionality is overly low, compressing the node factor $N$. This leads to the challenge of optimizing one-shot AE.
> >
> > **Q3.** In section 3.1 it is stated that graph matching is used for the AE training loss to ensure permutation and SE3 invariance. This can be computationally expensive. How fast is the AE training? In terms of wall time, but also asymptotically (in O notation)? Without knowing this the comparison of training time of the proposed method vs EDM is also a bit complicated. As I understand Table 4. does not account for the AE training time (yes, it can be trained once, but it still needs training).
> >
> > **Answer.**
> > - We train both topology and geometry AEs on one NVIDIA A100 GPU for 2 days. The latent diffusion training time depends on the datasets, e.g. in Geom-Drugs it takes ~4 hours compared to 5.5 days of EDM with the same epochs.
> > - Also noting that AEs are pretrained for once and repetitively utilized in varied scenarios e.g. conditional generation on varied properties. This is very important since effective and efficient multi-property optimization is highly demanded in the real-world drug discovery process (https://www.nature.com/articles/nrd1086).
> > - It is difficult to directly compare the asymptotic complexity. While both AE and EDM are composed of building blocks of invariant/equivalent representation neural networks, it would lead to similar asymptotic complexity if they are constructed with the same building blocks.
> >
> > **Q4.** Also, since AE is trained on a much larger dataset, it might be fair to also pre-train the EDM on general molecule generation. It has been shown that pretraining helps molecule generation (https://arxiv.org/pdf/2309.17296.pdf). While this paper is quite recent it would still make more sense to compare against such properly tuned state-of-the-art molecular diffusion setup instead of vanilla EDM, which is an older baseline (first ever diffusion model for 3D molecule generation).
> >
> > **Answer.**
> > - We thank the reviewer for bringing this interesting paper, though the numbers in the paper are not directly comparable to ours for its different setting, and we are currently unable to reproduce the carefully designed and complicated algorithm without their codebase public.
> > - The conclusion in the mentioned paper differs from us on pretraining. The mentioned paper concludes that the benefit of pretraining is in the small-data regime: “Interestingly, the fine-tuned model shares similar (best) scores with trained from scratch on 100% of the data” (Section 6). Our latent diffusion improves the generation quality in the large-data regime (100% of the data).
> > - We have implemented our diffusion model pretraining experiment in the larger PubChem3D database to validate the conclusion. The diffusion model pretraining is very time-consuming (compared to simply pretraining AE), expected to take more than 12 days of the rebuttal window. We will include the part of the experiment in our later revision.
> >
> > **Q5.** Why is MIDI, which is cited numerous times in the paper is not compared against in the experiments? It does in certain metrics perform a lot better than the EDM, which is used as the main baseline here.
> >
> > **Answer.**
> > - The reported results in MiDi were evaluated in a different setting as stated in Section 5.2, which we did not include before.
> > - We re-evaluated MiDi in the EDM setting and supplemented the result in Table 2.

---

> > > ### Author Response · Authors · 2023-11-21
> > > **Response to Reviewer 7uXM (3)**
> > >
> > > **Q6.** Also, in 3.2 Setup sections authors say that connectivity is commonly later determined based on domain rules and cite MIDI among others. Its true, that such domain-based rules are used to recover connectivity in e.g. EDM, but in MIDI connectivity is modeled explicitly if I remember correctly?
> > >
> > > **Answer.**
> > > - We are sorry for the confusion. The reviewer is correct that MiDi is mistakenly cited at this place. We removed it in the paper.

---

> ### Comment · Reviewer_7uXM · 2023-11-30
> **Response to the Rebuttal**
>
> Thank you for your answers.  I will increase my score accordingly. Although I'm still lukewarm about the significance of the contribution.

---

### Official Review · Reviewer_kaPc · 2023-11-06

**Soundness:** 2 fair
**Presentation:** 2 fair
**Contribution:** 2 fair
**Rating:** 5
**Confidence:** 3

**Summary:**

This paper proposes an algorithm for diffusion-based 3D graph generation through latent space. This paper claims that low-dimensional projection is the key factor of the graph generation due to satisfy the equivariant property. The experiments are conducted using 3D molecules datasets.

**Strengths:**

The writing is clear. The introduction section is wonderful. I clearly understand the scope of this work and what factors that the authors mainly addresses about. The meaning of the latent space and the 2D-3D autoencoder is well aligned in that the latent space could be understood as down-dimensional projection from 3D to 2D. Also, the experiments look okay and outperforms the previous works.

However, I am not sure of the clear contribution of this work. Let me write down my questions and worries in the following sections.

**Weaknesses:**

# W-1. Why latent space?

According to this paper, I did not clearly catch how the authors project the 3D molecules into the 2D latent space. The technical details are missing. It naively mentioned that the authors borrow the architectures from previous works and such simple comment is not self-contained, in my opinion.

However, let's assume that this issue is okay with me. Then, the following question is ... why the latent space is typically important to the graph generation? I know that the several recent studies utilize the latent space for the representation of the diffusion-based generative models. Nonetheless, I could not find any theoretical analysis behind the bridge between

- _'the necessity of the latent space'_  and
- _'the properties of the 3D graph representation, typically for the 3D molecules'_.

For me, this is naive extension of latent diffusion models into graph representation.

# W-2. Overclaim.

Can the authors exactly prove this equation in Sec 3 of this manuscript?

_3D Graph Diffusion Performance <= Latent Space Reconstruction Quality + Symmetry Preservation * Data Dimensionality_

 # W-3. Lack of experiments

If the main contributions of this paper is about the analysis of the latent space into the 3D graph representation, I think that the authors should have provided the clear results or ablation study about the this contribution.

For instance, using the same baseline model from (Hoogeboom et al., 2022), the authors could slightly modify the architecture while maintaining the number of parameters for fair comparison. However, I could not find such kind of analysis or experiment.

**Questions:**

Please address my concerns listed in Weakness section.

# Q-1. Proposition 1

While the authors describe full of equations with comments, the concept itself is highly straightforward. For instance in the manuscript, the authors said that '__Proposition 1.__ _Performance bound of 3D graph diffusion is related to feature dimensionality_'.
In my opinion, simply if we increase the network capacity by adding more layers or increasing the channel-length with lots of training data, the diffusion models surely have the performance gain. Honestly, I cannot catch the authors' intension from this propositions.

# Q-2. Ablation study

What if the authors intentionally omits the latent space encoding? I mean instead of using 2D-3D autoencoders, what happens if the network only consists of the 3D autoencoders? How much gain could we obtain if we adopt the latent space encoding?

# Q-3. Clear difference between the previous works.

Can the authors create one table to clearly demonstrate the difference? I read the paper (Hoogeboom et al., 2022) and it seems like this paper also split the geometry and topology for the diffusion process. Not just this factor itself, I hope to know the clear footsteps that this paper newly takes.

**Details Of Ethics Concerns:**

No ethics review required

---

> ### Author Response · Authors · 2023-11-21
> **Response to Reviewer kaPc (1)**
>
> **Q1.** According to this paper, I did not clearly catch how the authors project the 3D molecules into the 2D latent space. The technical details are missing. It naively mentioned that the authors borrow the architectures from previous works and such simple comment is not self-contained, in my opinion.
>
> **Answer.**
> - We are sincerely grateful to the reviewer’s feedback. We are committed to improving our clarity, and also would like to point out some aspects missed by the reviewers.
> - We add the detailed algorithm sections (Algorithms 1 & 2) to describe our latent 3D graph diffusion, beyond the overview Figures 2 & 3. They describe the processes of (conditional) in/equivariant latent encoding, latent diffusion, and sampling.
> - More specifically, the details of invariant latent encoding are further presented in Section 3.1.
> - The details of conditional in/equivariant latent encoding are presented in Section 3.3.
> Regarding latent encoding architectures, we supplement more detailed descriptions of architecture design in Appendix B.2 & C.1
> - The details of diffusion model are presented Section 2.
>
> **Q2.1.**  However, let's assume that this issue is okay with me. Then, the following question is ... why the latent space is typically important to the graph generation? I know that the several recent studies utilize the latent space for the representation of the diffusion-based generative models. Nonetheless, I could not find any theoretical analysis behind the bridge between
> 'the necessity of the latent space' and
> 'the properties of the 3D graph representation, typically for the 3D molecules'.
> For me, this is naive extension of latent diffusion models into graph representation.
>
> **Answer.**
> - **Our work is not a naive extension of latent diffusion into 3D graphs.** We aim to generate 3D graphs of symmetry-group equivariance through diffusion. In other words, the quality of the generated graphs of topology and geometry is invariant / equivariant to a set of transformations (permutation group and Lie group E(3) of rotations and translations in our study).  This aim has deep roots in 3D molecular generation as demonstrated in our numerical experiments for desired quantum properties (invariant to E(3) perturbation of the 3D molecules) or desired protein binding (equivariant to E(3) perturbation of the 3D molecules); and it has broader application scopes for 3D graphs of symmetry-group equivariance in general.
> - Thus, directly related to the reviewer’s question, **one contribution of our work is to demonstrate the necessity of building the 3D graph latent space** to achieve a potential generation quality. We provided theoretical analysis (Section 3.2) on what space should diffusion generative models operate on for generating 3D graphs of non-Euclidean structure and group-symmetry equivariance.  We addressed group-symmetry equivariance while deriving model performance bounds.
>     - Specifically, Proposition 1 gave the upper bound of the diffusion model performances (as measured in total variation distance) operating in the original 3D graph domain ($D’$-dimensional);
>     - whereas Proposition 2 gave the upper bound in the latent space ($D’’$-dimensional). The latter led to a better upper bound of the diffusion model performances if reconstruction errors are low and the latent dimension is low, which motivates our approach of latent diffusion.
> - **Another contribution is to illustrate how to construct such 3D graph latent space (autoencoder), guided by our theoretical, motivational analyses**, of the following properties: (1) permutation and E(3) invariant and (2) sufficiently low reconstruction errors despite low latent dimension (these two present nontrivial trade-offs), which is under-explored.  We accomplished this by combining 2D graph autoencoder (for topology features) and point cloud autoencoder (for geometry features given topologies) in a cascade manner (Section 3.1).
>
> **Q2.2.**  While the authors describe full of equations with comments, the concept itself is highly straightforward. For instance in the manuscript, the authors said that 'Proposition 1. Performance bound of 3D graph diffusion is related to feature dimensionality'. In my opinion, simply if we increase the network capacity by adding more layers or increasing the channel-length with lots of training data, the diffusion models surely have the performance gain. Honestly, I cannot catch the authors' intension from this propositions.
>
> **Answer.**
> - We are sorry for the confusion. The feature dimensionality in Proposition 1 refers to the feature dimensionality of data, which does not relate to the network capacity. We revised the comment for better clarity. Please also see our last answer for the intention behind Proposition 1.

---

> > ### Author Response · Authors · 2023-11-21
> > **Response to Reviewer kaPc (2)**
> >
> > **Q2.3.** Can the authors exactly prove this equation in Sec 3 of this manuscript?
> > 3D Graph Diffusion Performance <= Latent Space Reconstruction Quality + Symmetry Preservation * Data Dimensionality
> >
> > **Answer.**
> > - The plain language equation is the informal expression of Proposition 2, which is proved in Appendix A.
> >
> > **Q3.** Clear difference between the previous works.
> > Can the authors create one table to clearly demonstrate the difference? I read the paper (Hoogeboom et al., 2022) and it seems like this paper also split the geometry and topology for the diffusion process. Not just this factor itself, I hope to know the clear footsteps that this paper newly takes.
> >
> > **Answer.**
> > - We thank the reviewer for the good comments. Following it, we summarize the difference across different methods in Table 10.
> > - The table characterizes two perspectives of difference: (i) latent dimension and (ii) the modeling of topological and geometric features.
> >     - **Latent dimension (LD).** Assuming the feature dimension of data is $N \times (D+3)$. At the one end of the spectrum, EDM diffusing at the original space is of the maximum LD $N \times (D+3)$. The follow-up work, GeoLDM, diffusing at the latent space of nodes is of LD $N \times (D’+3)$. Guided by our theory in Section 3.2, we chase a lower dimensional latent space by compressing the node factor $N$, that our method at the other end of the spectrum of the minimum LD $D’$.
> >     - **Modeling of topological and geometric features.** EDM, GeoLDM and our one-shot model jointly model the topological and geometric features for all $N$ nodes. However, our one-shot model cannot manage to reconstruct the feature per node for its overly low LD, that the node factor $N$ is compressed. Guided by our theory in Section 3.2, we develop the cascaded model, achieving satisfactory reconstruction performance while maintaining symmetry and sufficiently low LD.
> >
> > **Q4.** If the main contributions of this paper is about the analysis of the latent space into the 3D graph representation, I think that the authors should have provided the clear results or ablation study about the this contribution.
> > For instance, using the same baseline model from (Hoogeboom et al., 2022), the authors could slightly modify the architecture while maintaining the number of parameters for fair comparison. However, I could not find such kind of analysis or experiment.
> >
> > What if the authors intentionally omits the latent space encoding? I mean instead of using 2D-3D autoencoders, what happens if the network only consists of the 3D autoencoders? How much gain could we obtain if we adopt the latent space encoding?
> >
> > **Answer.**
> > - The contribution of 3D graph latent spaces was demonstrated in our results, and the confusion may arise from a lack of a more explicit summary in our presentation. We added them in the paper. Following our answers in Q3:
> >     - Lower LD latent space, better generation quality: Shifting from EDM and GeoLDM by reducing LD per node, better generation quality is achieved.
> >     - Overly low LD latent space challenges its quality (in reconstruction) for generation: Shifting from GeoLDM to our one-shot model by further reducing the node factor in LD, it is difficult to achieve sufficient low reconstruction errors for generation.
> >     - More qualified latent space (in reconstruction), better generation quality: Shifting from our one-shot model to cascaded model, we achieve state-of-the-art generation results by building the 3D graph AE with low reconstruction errors.
> >     - More “well-structured” latent space (in semantics), better OOD generation quality: Shifting from our cascaded model to self-supervised cascaded model, we achieve better OOD conditional generation results when the homogeneity metric is higher.
> > - We also completely agree with the reviewer’s opinion of ablation studies, though the suggested one of slight variation from EDM is difficult to implement in the short period since our pipeline is not trivially extended, starting from EDM.
> > - We follow the suggestion to provide the result when the 3D latent space is omitted in Table 17, leading to a worse generation quality. Since the 2D latent space is the prerequisite for 3D in the cascaded model, it demonstrates the necessity of both AEs in our pipeline.

---

> > > ### Comment · Reviewer_kaPc · 2023-11-23
> > > **Thank you for the rebuttals.**
> > >
> > > Overall, I understand the author's analysis of the latent graph design. Also, the authors responded to most of my questions. However, I still feel like this is a naive extension of latent diffusion into a graph structure. While there are so many analyses in terms of this topic, there are not that many technical contributions that bridge the authors' analysis and their model design. I think that this is quite a controversial topic, however, as a reviewer, I am not that positive with this paper.
> > >
> > > Moreover, I still disagree with the authors' opinion on
> > >
> > > - _3D Graph Diffusion Performance <= Latent Space Reconstruction Quality + Symmetry Preservation * Data Dimensionality_
> > >
> > > While the authors said that there is an exact proof in Appendix A, I cannot find any conclusion that mathematically proves the exact equation that I am concerned about. Even though I can guess the authors' intentions, this is an overclaim. For me, once the authors write down the equation within the paper, the equation itself should be formed in an exact and self-contained manner. For me, this is not a general talk or a simple presentation. This is a theoretical paper. In my opinion, such an equation is not a theoretical claim, which makes me unsatisfactory with this submission.

---

> ### Comment · Reviewer_kaPc · 2023-11-23
> **Thank you for the rebuttals.**
>
> I thank the authors for their endeavor in preparing the rebuttal. Through the rebuttal, I can further understand the exact meaning and insights that the authors proposed. However, __I am not that satisfied with several issues: (1) lack of technical contributions, (2) overclaim in "The plain language equation"(Q3), and (3) naive extension of latent diffusion into a graph structure. For some issues, the authors provide their opinions. However, it looks trivial for me. Also, motivation itself is not that easy to understand.__
>
> Let me keep my score as _"5: marginally below the acceptance threshold"_. Though I am not that positive with this paper, if the other reviewers think this is okay, let me follow their opinions. However, the manuscript needs lots of modification and needs to deal with the exact problem and novelty. Though the authors updated the algorithm 1, I could hardly find clear insight from this paper.

---

> ### Author Response · Authors · 2023-11-23
> **Further Clarification to Contributions and Inequation**
>
> Dear Reviewer kaPc,
>
> We sincerely thank you again for your time and effort to review our paper and read our response. Here are our further clarifications to your questions:
>
> -------------------------------------
>
> - We respectfully disagree the (1) and (2) points of lacking technical novelty and lacking proof. We will illustrate this below.
> - **For (1) and (2)**: The inequation "*3D Graph Diffusion Performance <= Latent Space Reconstruction Quality + Symmetry Preservation * Data Dimensionality*" is **the informal expression of proposition 2**, which is **proved in Appendix A.2**. More specifically in proposition 2 (Eq. (3)):
>     - *3D Graph Diffusion Performance* denotes the left-hand-side of the inequation, measuring the statistical distance between learned distribution and data distribution;
>     - *Latent Space Reconstruction Quality* denotes the first term of the right-hand-side of the inequation, measuring the reconstruction error of the forward and backward mappings (i.e. reconstruction error of AE);
>     - *Symmetry Preservation* denotes the first multiplier of the second term of the right-hand-side of the inequation, measuring how AE preserves symmetry affecting the data-processing factor (see Appendix A.1 for details);
>     - *Data Dimensionality* denotes the second multiplier of the second term of the right-hand-side of the inequation, proportional to the dimension of data.
>
>
> -------------------------------------
>
> - We respectfully disagree for **the point (3)** of lacking novelty. Our innovations lie in four aspects, which are detailed in previous responses and briefly summarized below,
>     - Contribution (i) is exactly the technical rationale as we responded (Sec. 3.2);
>     - Contribution (ii) is also guided by the analysis, of how to build a qualified 3D graph AE (Sec. 3.1);
>     - Contribution (iii) is the extension of our latent diffusion pipeline to the **in/equivariant conditional generation** setting (Sec. 3.3). This is an entirely new exploration in the field;
>     - Contribution (vi) is the exploration of graph self-supervised learning for graph generative models, to improve the generalizability of graph generative AI (Sec. 3.1).
>
> -------------------------------------
>
> We completely respect the dispute here, and also hope that our clarification will assist the reviewer in addressing the confusion more effectively.

---

### Official Review · Reviewer_oqpy · 2023-11-08

**Soundness:** 3 good
**Presentation:** 3 good
**Contribution:** 3 good
**Rating:** 6
**Confidence:** 3

**Summary:**

This study investigated the generation of 3D graphs from latent space using diffusion models and conditioned on different properties. The model is based on a cascaded 2D-3D graph autoencoders combined with diffusion models. The authors investigated the usefulness of generating 3D graphs from the latent space with symmetry preserved, and explored latent diffusion for conditional 3D graph generation.

**Strengths:**

The paper is well written and the rational is straightforward and convincing. The authors have conducted multiple numerical experiments and the conditional generation based on various properties show great potentials of such method. In general I find this is an interesting paper.

**Weaknesses:**

Please see my questions below.

**Questions:**

The authors trained the topological AE and the geometric AE separately with different constrains. I understand the difficulty in training them in one shot, but I'm wondering the influences from each of the AE, e.g., would it be possible information about the topology is lost while training using the geometric AE? If so, how much is lost/kept?

While GSSL improves the results on OOD, it seems it can worse the results on ID. Can the authors provide more insights on this?

Considering one utility of the model is for generating new drugs, model interpretability could be important. In the latent space, would it be possible for the authors to provide some visualization on the learned features and if available, colored by some topological and geometric features of the graph? If there are certain patterns there then it may be useful to help better understand what features are mostly kept in the latent space and enhance the interpretability of the model.

---

> ### Author Response · Authors · 2023-11-21
> **Response to Reviewer oqpy**
>
> **Q1.** The authors trained the topological AE and the geometric AE separately with different constrains. I understand the difficulty in training them in one shot, but I'm wondering the influences from each of the AE, e.g., would it be possible information about the topology is lost while training using the geometric AE? If so, how much is lost/kept?
>
> **Answer.**
> - In the cascaded model, the geometric AE still preserves topology information, and the topological AE does not have access to geometry information.
> - Specifically, the model (i) performs autoencoding on topological features, and then (ii) given the topological features as inputs, performs autoencoding on geometric features.
> - The information loss in the topological AE is minor to our pipeline, as quantified with the reconstruction performance shown in Table 1 and Appendix B.2.
>
> **Q2.** While GSSL improves the results on OOD, it seems it can worsen the results on ID. Can the authors provide more insights on this?
>
> **Answer.**
> - We provide additional results in Tables 15 & 16, on applying GSSL for unconditional generation and conditional generation with less training data.
> - Based on our experiments, we observed that the benefit of GSSL is mainly on:
>     - OOD scenarios when the GSSL prior is aligned with the condition semantics, as we stated in Section 4.2;
>     - Unconditional or ID scenarios when training data are insufficient. This is consistent with the role of GSSL in discriminative modeling (https://arxiv.org/abs/2010.13902).
> - For the OOD scenarios, though the improvement does not appear for all datasets, we know when it is improved: Specifically, we can measure the semantics-alignment by computing the homogeneity ratio as detailed in Appendix E.2. This indicates a future potential to design semantics-specific GSSL tasks for generative modeling.
>
> **Q3.** Considering one utility of the model is for generating new drugs, model interpretability could be important. In the latent space, would it be possible for the authors to provide some visualization on the learned features and if available, colored by some topological and geometric features of the graph? If there are certain patterns there then it may be useful to help better understand what features are mostly kept in the latent space and enhance the interpretability of the model.
>
> **Answer.**
> - We provided the latent embedding visualization in Figures 11 & 12.
> - Specifically, we perform t-SNE (https://jmlr.org/papers/v9/vandermaaten08a.html) on topological and geometric embeddings, and annotate them with four 2D and six 3D property values of molecules.
> - Our visualization shows information of 2D properties is more preserved in topological embeddings, that data with similar properties tends to cluster. In addition, information of 3D properties is more preserved in geometric embeddings.
> - The observation is also confirmed by quantitative evaluation, by computing the silhouette score of the latent embeddings w.r.t. properties.

---

### Official Review · Reviewer_N14e · 2023-11-13

**Soundness:** 3 good
**Presentation:** 3 good
**Contribution:** 3 good
**Rating:** 8
**Confidence:** 4

**Summary:**

This paper investigates the appropriate latent space for generating 3D graphs. The authors derive several conclusions through theoretical analysis:

- The lower the dimensionality of the latent space, the higher the performance limit of diffusion.

- Higher quality of the latent space (i.e., lower reconstruction error) corresponds to a higher performance limit of diffusion.

- Preserving symmetry (maintaining graph properties after translation, rotation, etc.) is an inductive bias of the latent space, contributing to an increased performance limit of diffusion.

Guided by these theoretical insights, the authors propose a method named "latent 3D graph diffusion." This approach utilizes cascaded 2D-3D graph autoencoders to learn a latent space with low-error reconstruction (learning topological graphs) and symmetry invariance (learning geometric graphs). Furthermore, the authors extend this method to conditional generation given SE(3) invariant properties (rotation-translation invariance) or equivariant 3D objects.

Experimental results demonstrate that appropriate regularization of the latent space through graph self-supervised learning can further enhance the robustness of conditional generation. The comprehensive experimental findings indicate that, compared to existing competitive methods, this approach can generate 3D molecular graphs with enhanced effectiveness/drug similarity and is at least an order of magnitude faster in diffusion training. The speed advantage increases with the size/complexity of molecules.

**Strengths:**

- Demonstrates superior generation quality, rapid generation capabilities, and conditional generation proficiency in 3D graph generation, while enhancing robustness through regularization.

**Weaknesses:**

- Limited generalization capability in comparison.

**Questions:**

- How to improve generalization capability?

---

> ### Author Response · Authors · 2023-11-21
> **Response to Reviewer N14e**
>
> **Q1.** Limited generalization capability in comparison. How to improve generalization capability?
>
> **Answer.**
> - We provide additional results in Tables 15 & 16, on applying GSSL for unconditional generation and conditional generation with less training data.
> - Based on our experiments, we observed that the benefit of GSSL is mainly on:
>     - OOD scenarios when the GSSL prior is aligned with the condition semantics, as we stated in Section 4.2;
>     - Unconditional or ID scenarios when training data are insufficient. This is consistent with the role of GSSL in discriminative modeling (https://arxiv.org/abs/2010.13902).
> - Our results indicate a future potential to design semantics-specific GSSL tasks for generative modeling.

---

### Author Response · Authors · 2023-11-22
**Gentle Reminder**

Dear reviewers,

Thank you all again for your comments that have helped our revision. We hope that you have found our responses useful and our revision satisfactory. We would be thrilled to have more such exciting and constructive discussions with you. Could you please kindly share any comments to our responses at your earliest convenience so that we could still respond before the author-reviewer discussion window closes on November 22? Or if you have already found our responses satisfactory, we humbly remind you of a fitting update of the rating. Thank you all again for your time and efforts!

Sincerely, All authors

---

### Meta-Review · Area_Chair_7kgh · 2023-12-07

**Metareview:**

This paper studies graph generation by introducing a latent 3D graph diffusion framework. The approach involves training an autoencoder on graph data and constructing a diffusion model within the latent space of the trained autoencoder. Through experiments in drug discovery, the paper demonstrates the superiority of the proposed pipeline in both generation quality and diffusion training efficiency. While the paper addresses an interesting and essential problem in AI for science, its novelty is considered marginal. The proposed pipeline appears straightforward and lacks a significant contribution. The rebuttal has tackled many concerns raised by the reviewers, but some issues regarding marginal technical contributions and the need for improved presentation persist. Based on SAC's recommendation, the decision is to accept the paper.

**Justification For Why Not Higher Score:**

Lack of novelty and lack of clarity in presentation are major issues.

**Justification For Why Not Lower Score:**

NA

---

### Decision · Program_Chairs · 2024-01-16

Accept (poster)